



# Flywheel calibration of a continuous-wave coherent Doppler wind lidar

Anders Tegtmeier Pedersen[1] and Michael Courtney[1]

[1]Technical University of Denmark – DTU Wind Energy, Frederiksborgvej 399, 4000 Roskilde, Denmark

**Correspondence:** Anders Tegtmeier Pedersen (antp@dtu.dk)

**Abstract.** A rig for calibrating a continuous-wave coherent Doppler wind lidar has been constructed. The rig consists of a rotating flywheel on a frame together with an adjustable lidar telescope. The laser beam points toward the rim of the wheel in a plane perpendicular to the wheel's rotation axis, and it can be tilted up and down along the wheel periphery and thereby measure different projections of the tangential speed. The angular speed of the wheel is measured using a high-precision measuring ring

fitted to the periphery of the wheel and synchronously logged together with the lidar speed. A simple, geometrical model shows that there is a linear relationship between the measured line-of-sight speed and the beam tilt angle and this is utilised to extrapolate to the tangential speed as measured by the lidar. An analysis of the uncertainties based on the model shows that a standard uncertainty on the measurement of about $0.1\%$ can be achieved, but also that the main source of uncertainty is the width of the laser beam and it's associated uncertainty. Measurements performed with different beam widths confirms this.

Other measurements with a minimised beam radius shows that the method in this case performs about equally well for all the tested reference speeds ranging from about $3\,\mathrm{m/s}$ to $18\,\mathrm{m/s}$.

## 1   Introduction

Wind lidars are often referred to as being 'absolute' instruments by which is meant that, given only the two parameters; the

laser wavelength and the frequency at which we sample the backscattered light, we are able to calculate the measured line-of-sight (LOS) speed through the well-known equation $V = \frac{1}{2}\lambda \cdot \Delta f$, Pearson et al. (2002). This to some implies that wind lidars are also 'calibration free' since there, in contrast to e.g. cup anemometers, are no empirical constants to be found through a calibration. However, a calibration is fundamentally just a comparison to a reference with a known and traceable uncertainty (Joint Committee for Guides in Metrology (2012)), and without a calibration we have no way of knowing that the lidar measures

correctly; small errors can easily creep into the frequency analysis or the laser wavelength may drift etc. Equally important, by using a reference with known uncertainty traceable to international measurement prototypes, we can assign an uncertainty to the lidar radial speed and claim traceability. The latter is often a requirement in commercial measurements where the outcome can have financial consequences.





Therefore, and inspired by a similar concept commonly used for calibrating Laser Doppler Anemometers (LDAs), Shinder et al. (2013), we have constructed a rig for calibrating Doppler lidars. The rig in essence consists of a frame with a stainless-steel flywheel in one end and an adjustable lidar telescope pointing toward the wheel rim in the other. However, conversely to an LDA system a Doppler lidar measures the velocity component along the laser beam and we therefore use the lidar

beam skimming on the circumference rather than impinging the wheel surface perpendicularly as the LDA would do. The telescope is mounted on a pivoting mechanism and with this the laser beam can be tilted and thus different projections of the wheel peripheral speed probed. In addition, we have developed a simple model relating the ratio between the speed sensed by the lidar and the peripheral speed to the beam tilt angle and this method allows us to estimate the true peripheral speed by extrapolation from speeds measured at other angles.

This manuscript is organised in the following way. First the calibration rig and lidar are described. Then the model describing the relation between the measured line-of-sight speed and tilt angle is gradually developed beginning from a simple 2D model to a more realistic 3D model. The model forms the basis of the following suggestion for a calibration procedure and analysis of the various uncertainty contributions

## 2   Calibration rig and lidar

The calibration rig consists of an aluminium frame on which is mounted a stainless steel wheel together with the transceiver, or telescope, of the lidar. The wheel has a radius of 286.76 mm with a measured eccentricity of about 0.01 mm and it is coupled directly to a servo motor to control its rotational speed. The telescope is mounted at approximately the same height as the top of the wheel and in such a way that it can be tilted around a horizontal axis parallel to the wheel's rotational axis using a fine threaded adjustment screw. On top of the telescope is mounted an inclinometer to measure the tilt angle of the laser beam, see

Fig. 1.

In order to measure the rotational speed, a high-precision measuring ring is fitted to the periphery of the wheel together with a corresponding measurement head sitting near the bottom of the wheel, AMO GmbH (2013). This measurement system is in essence a transformer with a moving reluctance core; the ring is engraved with reluctance graduations at a pitch of $1000\,\mu$m with a precision of $\pm3\,\mu$m which induces voltages in the static measurement head as the wheel rotates. The output of the

system is 1800 TTL pulses per wheel revolution, and the period for six consecutive pulses are measured, and inverted to give the wheel rotational frequency.

The lidar is a direction sensitive continuous-wave coherent Doppler lidar operated with a 1565 nm fibre laser, Pedersen et al. (2014). The Doppler spectra are based on a 1024 point discrete Fourier transform (DFT) of the detector output sampled at 120 MHz resulting in a spectrum resolution of 117 kHz or 0.0917 m/s. About 1200 spectra are combined to form one average

spectrum at a rate of approximately 100 Hz. Based on the average spectrum the radial speed is estimated as the 50% fractile of the signal exceeding the detection threshold, Angelou et al. (2012). Focusing of the laser beam is controlled by adjusting the distance from the laser output fibre to the focusing lens with a micrometer screw. The lens has a 1" diameter and a focal length of 0.10 m. Laser, telescope and detectors are connected by optical fibres.





**Table 1.** Physical properties of calibration rig and lidar.

| | | | |
|---|---|---|---|
| Wheel radius | $R$ | [mm] | 286.76 |
| Wheel eccentricity | $e$ | [mm] | 0.01 |
| Distance telescope to wheel | L | [m] | 1.578 |
| Encoder pulses | | [pulse/rev] | 1800 |
| Encoder pitch | | [$\mu$m] | 1000$\pm$3 |
| Lens diameter | m | 0.0254 | |
| Lens focal length | m | 0.10 | |
| Laser wavelength | $\lambda$ | [nm] | 1565 |
| Sampling rate detector output | $f_s$ | [MHz] | 120 |
| Number of points in DFT | NDFT | | 1024 |
| Bin width | | [m/s] | 0.0917 |
| Measurement rate | Hz | 100 | |

Real-time signals from lidar, inclinometer, and rotation encoder are streamed to a measurement computer which synchronises at 100 Hz and stores the data for post processing.

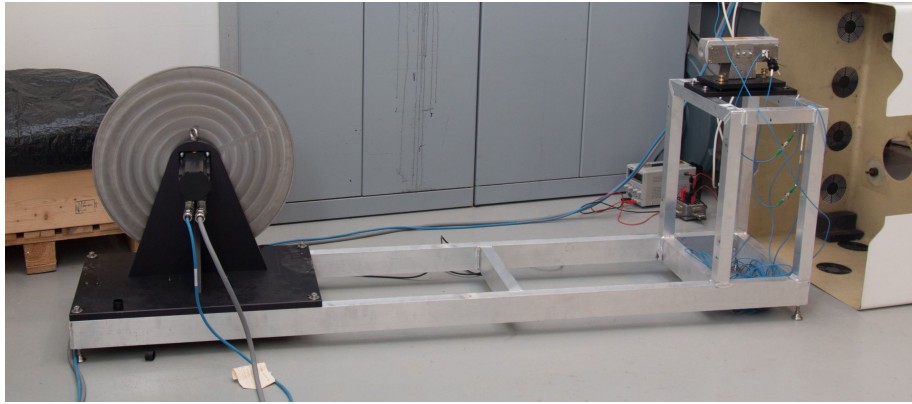

**Figure 1.** Photo of the calibration rig. To the left is seen the flywheel with cables through which the motor is controlled and to the right the lidar telescope with optical cables connecting it with the laser and detectors.

## 3 Model and calibration procedure

Using a rotating wheel for calibration has been used with LDAs for many years, Shinder et al. (2013); Bean and Hall (1999);
5 Duncan and Keck (2009), and as mentioned in Sect. 1 our calibration rig is strongly inspired by what has been done with





LDAs. However, unlike LDAs coherent Doppler lidars measure the velocity component along the laser beam meaning that the beam must be aligned with and overlapping a tangent of the wheel and this poses a severe problem. For the lidar to measure the true tangential wheel speed, and only that, the overlap between laser beam and wheel surface will need to be infinitely small and there will be no backscatter signal to detect. It is therefore necessary to measure a different component of the reference

speed than the tangential together with the corresponding tilt angle and from that calculate the measured tangential speed. Unfortunately, this approach also has some drawbacks in the form of additional uncertainties due to the tilt angle measurement. Another source of uncertainty, present at any tilt angle, is the speed estimation uncertainty due to the finite resolution of the measured Doppler spectrum; this can however be eliminated by scanning over a range of tilt angles or alternatively, a range of wheel speeds. We have developed a model from simple geometric considerations describing the ratio between the tangential

wheel speed and the speed sensed by the lidar as function of beam tilt angle. The model shows that this relationship is linear and it can therefore be used to make a simple extrapolation back to what would be the tangential wheel speed sensed by the lidar.

    In the following subsections the model is derived; first under the approximation that the laser beam has no transverse component (i.e. it is infinitely narrow), during which we establish the relationship between the beam tilt angle, $\theta$, and the

skimming angle, $\phi_s$, and later for a collimated beam of finite width, $w$. Finally, the suggested procedure for doing the calibration is described.

### 3.1   Infinitely narrow beam

Figure 2 shows a schematic drawing of the calibration setup. The line-of-sight speed sensed by the lidar ($V_{LOS}$) is the wheel's peripheral velocity projected into the direction of the beam at the point of intersection between the wheel surface and laser

beam. From Fig. 2 this is seen to be

$$V_{LOS} = V_{wheel} \cos(\phi_s + \theta) = \omega R \cos(\phi_s + \theta), \tag{1}$$

where $V_{wheel}$ is the peripheral speed, $\phi_s$ is the skimming angle, $\theta$ is the beam tilt angle, $\omega$ the angular frequency, and $R$ the radius of the wheel.

    Now, to find the relation between $\phi_s$ and $\theta$ we can make use of a little trick which will also prove valuable later on; instead

of tilting the beam we rotate the centre of the wheel, $(x_0, y_0)$, an angle $\theta$ around the centre of the transceiver lens which defines the origo of our coordinate system, see Fig. 3. The new centre of the wheel is denoted $(x_r, y_r)$. From the figure it is clear that the angle, $\phi_r$, between the vertical and the intersection point between beam and wheel is

$$\phi_r = \phi_s + \theta, \tag{2}$$

and that

$$\cos(\phi_r) = \frac{-y_r}{R}. \tag{3}$$





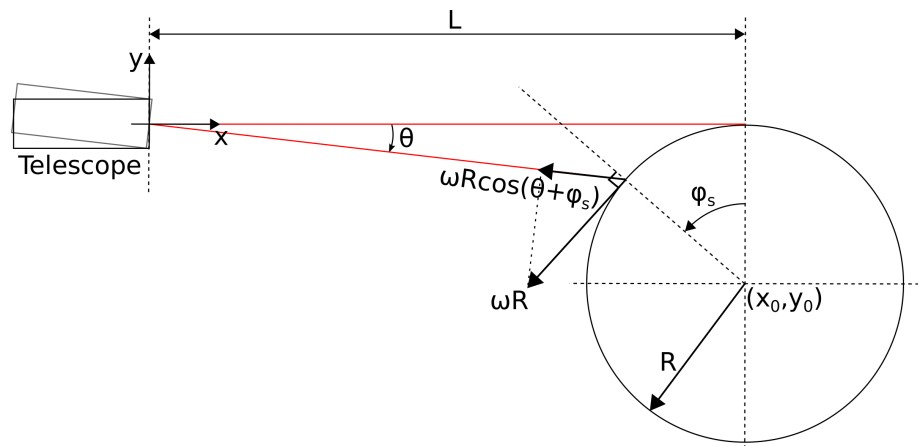

**Figure 2.** Schematic drawing of the calibration rig illustrating the basic geometry of the rig. The laser beam, illustrated in red, can be tilted using an adjustment screw on the telescope mount.

From Fig. 2 we see that $(x_0, y_0) = (L, -R)$ and from this we can calculate $(x_r, y_r)$ via the rotation matrix $\underline{\underline{R}}_z(\theta)$

$$\begin{pmatrix} x_r \\ y_r \end{pmatrix} = \underline{\underline{R}}_z(\theta) \begin{pmatrix} x_0 \\ y_0 \end{pmatrix} = \begin{pmatrix} \cos\theta & -\sin\theta \\ \sin\theta & \cos\theta \end{pmatrix} \begin{pmatrix} L \\ -R \end{pmatrix} = \begin{pmatrix} L\cos\theta + R\sin\theta \\ L\sin\theta - R\cos\theta \end{pmatrix}. \tag{4}$$

Now, rearranging and inserting into Eq. (1) we finally arrive at

$$\frac{V_{\mathrm{LOS}}}{V_{\mathrm{wheel}}} = \frac{R\cos\theta - L\sin\theta}{R}. \tag{5}$$

In our setup $\theta$ is small reaching a maximum value of about 2.5° so we can make the approximations that $\cos\theta = 1$ and $\sin\theta = \theta$ such that

$$\frac{V_{\mathrm{LOS}}}{V_{\mathrm{wheel}}} \approx 1 - \frac{L\theta}{R}. \tag{6}$$

It is thus seen that for small tilt angles there is a linear relationship between $V_{\mathrm{LOS}}/V_{\mathrm{wheel}}$ and $\theta$ and this can be utilised in the
10 calibration procedure.

## 3.2 Finite width, collimated beam, 2D

Now, a real laser beam is of course not infinitely narrow but has a transverse profile of finite width, e.g. the laser used in this study has a Gaussian profile. We therefore expand the model to include the beam width radius, $w$, but to begin with limit our selves to the two dimensional case and the assumption that the beam intensity has a constant transverse cross-section, i.e. the
15 intensity profile across the beam has a "top hat shape".




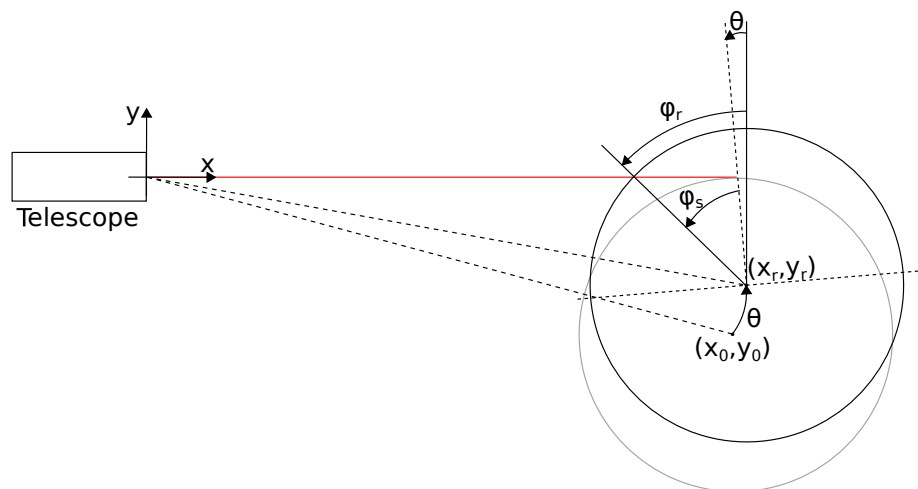

**Figure 3.** Instead of tilting the beam the centre of the wheel is rotated around the lens centre.

For a beam of finite width a finite part of the wheel perimeter will be illuminated by the laser and thus a range of line-of-sight speeds be measured, see Fig. 4. Each incremental line-of-sight speed will be

$$dV_{\mathrm{LOS}} = V_{\mathrm{wheel}} \cos\left(\phi_r\right), \tag{7}$$

and these will each contribute a proportion $d\phi_r/\Delta\phi_r$ of the total speed sensed by the lidar, where $\Delta\phi_r$ $(= \phi_{r_1} - \phi_{r_0})$ is the
total angle subtended by lidar illumination. The total speed contribution $V_{\mathrm{LOS}}$ is thus obtained by integrating Eq. (7) whilst normalising by $\Delta\phi_r$

$$V_{\mathrm{LOS}} = \frac{1}{\Delta\phi_r} \int_{\phi_{r_0}}^{\phi_{r_1}} V_{\mathrm{wheel}} \cos\left(\phi + \theta\right) d\phi = \frac{1}{\Delta\phi_r} V_{\mathrm{wheel}} \left(\sin\phi_{r_1} - \sin\phi_{r_0}\right). \tag{8}$$

By applying L'Hospital's rule this is easily seen to reduce to $\cos\phi_r$ as $\phi_{r_1}$ approaches $\phi_{r_0}$, i.e. as the beam becomes narrower, and therefore give the same result as in Sect.3.1.
If we apply Taylor's expansion to the third order to Eq. (8) we get

$$
\begin{aligned}
\frac{V_{\mathrm{LOS}}}{V_{\mathrm{wheel}}} &= & \frac{1}{\Delta\phi}\left(\phi_1 - \frac{1}{6}\phi_1^3 - \phi_0 + \frac{1}{6}\phi_0^3\right) \\
&= & \frac{1}{\Delta\phi}\left(\Delta\phi - \frac{1}{6}\left(\phi_1^3 - \phi_0^3\right)\right) \\
&= & 1 - \frac{1}{6}(\phi_1^2 + \phi_0^2 + \phi_1\phi_0),
\end{aligned}
\tag{9}
$$

and if we further make the approximations

$$\phi_0 = \phi_{\mathrm{m}} - \delta,$$

$$\phi_1 = \phi_{\mathrm{m}} + \delta,$$





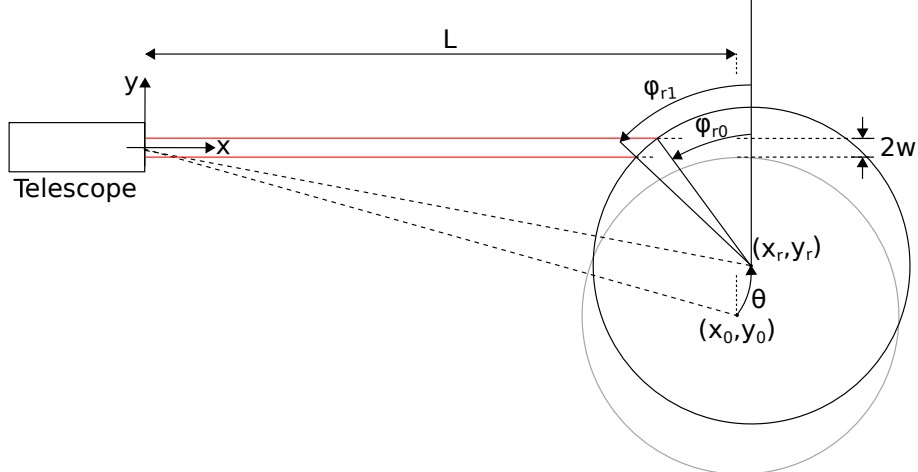

**Figure 4.** Schematic drawing used to derive the 2D thick beam model. Notice that $w$ is the beam radius and $(x_0, y_0 = L, -R - w)$.

where $\phi_{r_\mathrm{m}}$ is the mean of $\phi_{r_0}$ and $\phi_{r_1}$ and $\delta$ is a small perturbation we get

$$\frac{V_{\mathrm{LOS}}}{V_{\mathrm{wheel}}} = 1 - \frac{1}{6}(\phi_1^2 + \phi_0^2 + \phi_1\phi_0) = \qquad\qquad 1 - \frac{1}{6}\left(3\phi_\mathrm{m}^2 + \delta^2\right) \approx 1 - \frac{1}{2}\phi_\mathrm{m}^2 \approx \cos\phi_{r_\mathrm{m}}, \tag{10}$$

which is seen to be equal to Eq. (6).

This means that even for a beam of finite width Eq. (6) is a reasonable approximation to how the ratio $\frac{V_{\mathrm{LOS}}}{V_{\mathrm{Wheel}}}$ changes as
the beam is tilted. This can from a physical point of view intuitively be understood as that the high speed measured at $\phi_1$ is more or less balanced by the low speed measured at $\phi_0$. However, the approximation only applies as long as the entire beam cross-section is on the wheel; if part of the beam goes above the wheel, as it will for very small tilt angles, the relationship changes as we shall see in Sect. 3.2.1.

### 3.2.1 Special case: $\phi_0 = 0$

As mentioned above Eq. (6) only applies as long as all of the beam is on the wheel and it is therefore interesting to take a closer look at the special case when $\phi_0 = 0$, that is for so small tilt angles that parts of the beam go above the wheel. In that case Eq. (8) reduces to

$$\frac{V_{\mathrm{LOS}}}{V_{\mathrm{wheel}}} = \frac{1}{\phi_{r_1}}\left(\sin(\phi_{r_1})\right), \tag{11}$$

and if we Taylor expand we get

$$\frac{V_{\mathrm{LOS}}}{V_{\mathrm{wheel}}} = 1 - \frac{1}{6}\phi_{r_1}^2. \tag{12}$$

This means that as long as only a part of the beam is impinging on the wheel the sensitivity to the tilt angle is only a third compared to when the entire beam illuminates the wheel. The range of angles to which this applies obviously depends on the




beam width. The angle where the top of the beam first touches the wheel, i.e. when the entire beam is on the wheel, is denoted $\theta_1$ and is calculated through

$$\theta_1 = \arctan\left(\frac{2w}{L}\right). \tag{13}$$

### 3.3 3D model

In the previous section we modelled the laser beam intensity profile as a 2D top-hat shape. This is in conflict with the physical reality in two ways; firstly, confining the model to two dimensions effectively means that we are assuming the beam cross-section to be square and not round and secondly the real laser beam has a Gaussian intensity profile and not a top-hap shape. To take these facts into account we must therefore expand the model to three dimensions.

Still assuming that the beam is collimated we can model the beam as a cylinder of radius $w$ centred around the $x$-axis

$$y^2 + z^2 = w^2, \tag{14}$$

and the wheel as a cylinder along the $z$-axis and centred around $(x_r, y_r)$

$$(x - x_r)^2 + (y - y_r)^2 = R^2. \tag{15}$$

The $x$-coordinates of the overlap between beam and wheel in the rotated frame of reference is found by solving Eq. (15)

$$x = -\sqrt{R^2 - (y - y_r)^2} + x_r, \tag{16}$$

where $y \leq R + y_r$ and the sign of the square root is chosen such that only parts of the wheel facing the telescope are illuminated. The corresponding $y$ and $z$-coordinates are governed by Eq. (14) such that $(y_r, z_r) = (y, z)$. It should be noticed that in this way the overlap between wheel and beam has been parametrised into a function of $y$ and $z$ i.e. $g(x, y, z) = g(X(y), y, z)$.

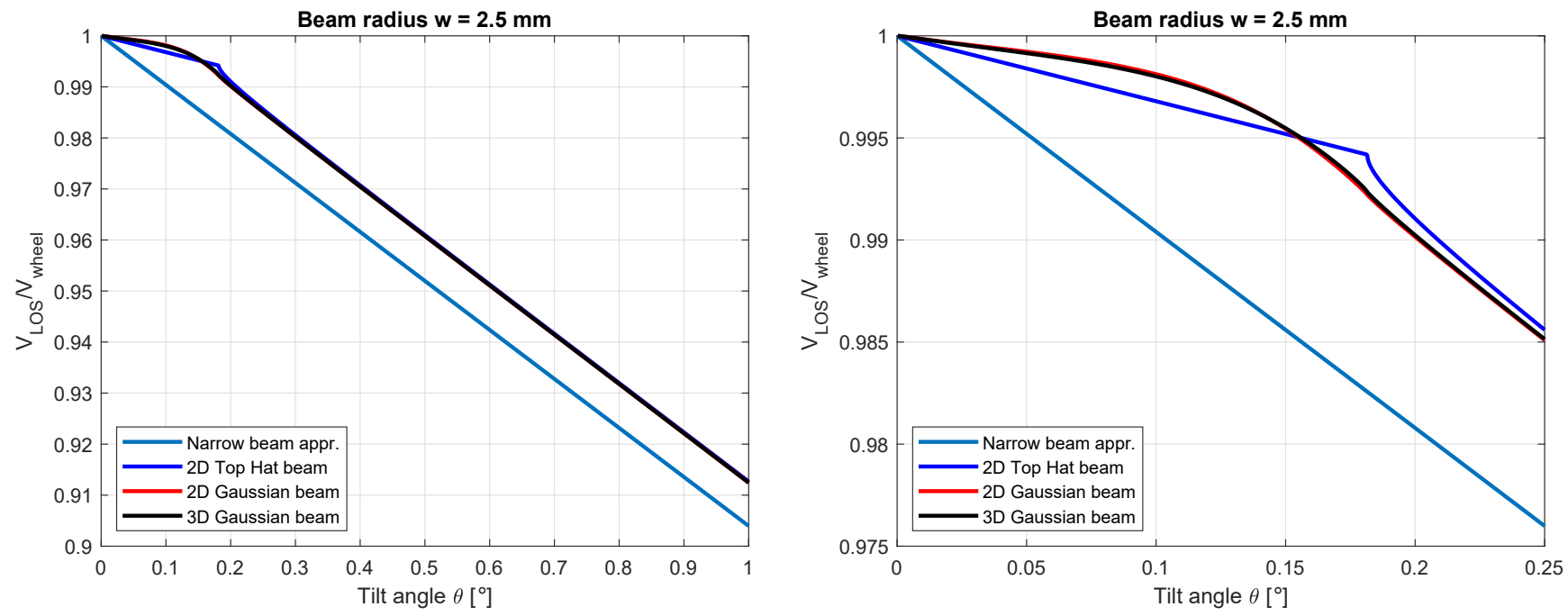

**Figure 5.** Comparison of the different models evaluated for tilt angles from $0-1^\circ$ to the left and to the right a close-up focusing on the shallow angles where only a part of the beam touches the wheel. For the chosen beam radius $\theta_1 = 0.18^\circ$.

In order to find the ratio $\frac{V_{\text{LOS}}}{V_{\text{Wheel}}}$ we follow the same procedure as in Sect. 3.2 by integrating all the speed contributions and normalise by the area of the illuminated surface, $\mathcal{S}$. This can be done by calculating the surface integrals

$$\begin{aligned}
\frac{V_{\text{LOS}}}{V_{\text{Ref}}} &= \frac{\iint\limits_{\mathcal{S}} I(y,z)\cos\phi \,\mathrm{d}S}{Ar(\mathcal{S})} \\
&= \frac{\iint I(y,z)\cos\phi\sqrt{1+\left(\frac{\partial X(y)}{\partial y}\right)^2+\left(\frac{\partial X(y)}{\partial z}\right)^2}\mathrm{d}y\mathrm{d}z}{\iint\sqrt{1+\left(\frac{\partial X(y)}{\partial y}\right)^2+\left(\frac{\partial X(y)}{\partial z}\right)^2}\mathrm{d}y\mathrm{d}z} \\
&= \frac{\iint I(y,z)\frac{y-y_r}{R}\sqrt{1+\left(\frac{\partial X(y)}{\partial y}\right)^2+\left(\frac{\partial X(y)}{\partial z}\right)^2}\mathrm{d}y\mathrm{d}z}{\iint\sqrt{1+\left(\frac{\partial X(y)}{\partial y}\right)^2+\left(\frac{\partial X(y)}{\partial z}\right)^2}\mathrm{d}y\mathrm{d}z} \\
&= \frac{\iint I(y,z)\frac{y-y_r}{R}\sqrt{1+\left(\frac{y-y_r}{\sqrt{R^2-(y-y_r)^2}}\right)^2}\mathrm{d}y\mathrm{d}z}{\iint\sqrt{1+\left(\frac{y-y_r}{\sqrt{R^2-(y-y_r)^2}}\right)^2}\mathrm{d}y\mathrm{d}z} \\
&= \frac{\iint I(y,z)\frac{y-y_r}{\sqrt{R^2-(y-y_r)^2}}\mathrm{d}y\mathrm{d}z}{\iint\frac{R}{\sqrt{R^2-(y-y_r)^2}}\mathrm{d}y\mathrm{d}z},
\end{aligned} \tag{17}$$

where $I(y,z)$ is the beam intensity profile.

### 3.4 Model comparison

In order to compare the different models from the simple narrow beam approximation and 2D top hat beam to the full 3D Gaussian beam a numerical evaluation of each has been performed for a beam radius of 2.5 mm ($1/e^2$-radius for the Gaussian profiles) and plotted together as function of tilt angle in Fig. 5. The left plot shows the models evaluated from $\theta = 0-1^\circ$ and





the right is a close-up focusing on the transition range where more and more of the beam falls on the wheel. The values used for $R$ and $L$ are the same as for the actual calibration rig.

Starting from angles larger than $\theta_1$ we see that all the models fall off with the same slope as predicted by the narrow beam approximation. Again, this indicates that as long the entire beam is on the wheel the sensitivity to a change in tilt angle is the
same for all beam widths and we can use Eq. (6) to calculate this sensitivity. On the other hand it is also clear that the beam width introduces an offset between the narrow beam and finite width models such that the wider beam measures a slightly higher speed than the narrow. This can be seen as the upper and lower parts of the beam not balancing each other perfectly; because of the curvature of the wheel the upper part spreads over a wider part of the wheel and therefore a wider range of speeds. This means that the absolute lidar measurement for a given tilt angle depends on the beam width, and it is therefore
critical to know this.

For angles smaller than $\theta_1$ the models stand out more clearly from each other. The 2D beam with a top hat transverse profile drops linearly from $\theta = 0°$ to $\theta_1$ where there is an abrupt change followed by a non-linear change in slope but soon it tends toward slope of the narrow beam. This abrupt change is due the discontinuous nature of the assumed beam profile. It should be noted though that due to the abrupt change the Taylor expansions in Eqs. (6) and (12) do actually not meet in $\theta_1$. The two beams
with a Gaussian profile behave differently with a smooth transition from the two regimes because near the edges of the beam the laser intensity is lower and therefore contributes less to the individual measurement. It is interesting to see how similar the two Gaussian models behave indicating that including the third dimension is not critical as long as a Gaussian transverse profile is used.

### 3.5    Wheel eccentricity

As written in Sect. 2 the wheel when mounted on the servo has an eccentricity of about $0.01$ mm. This eccentricity may come from either the wheel itself or from the mounting on the motor so that the wheel centre and centre of rotation is not perfectly aligned. In order to model the eccentricity and its effect on the calibration we will here assume the latter meaning that we model the wheel as being ideal but with it's centre off-sat from the centre of rotation by the amount $e$. Furthermore, we limit ourselves to regard the beam as having no transverse extent as we did in Sect. 3.1.

Figure 6 shows a schematic drawing of the situation adopting the method of rotating the centre of rotation around the lens. The wheel rotates around the point $(x_r, y_r)$ and the centre of the wheel, $c_w$, therefore follows a circle of radius $e$ around it. The tangential speed at the intersection between wheel and beam is proportional to the distance, $R_e$, from the rotation centre to the intersection point and the proportionality constant is of course the angular velocity, $\omega$. $R_e$ is a function of the rotation angle $\psi$. The lidar measures the projection of the tangential speed onto the laser beam and is thus given by $V_{\text{LOS}} = \omega R_e(\psi)\cos\phi_r$.
From the drawing we can see that that

$$\cos\phi_r = \frac{-y_r}{R_e(\psi)} \Rightarrow V_{\text{LOS}} = \omega R_e(\psi)\frac{-y_r}{R_e(\psi)} = -\omega y_r. \tag{18}$$

This means that the dependence on $\psi$ disappears and the measured line-of-sight speed will be the same for all rotation angles. Now, as the wheel rotates it can happen that the intersection lies to the right of the point $x_r$, as exemplified by the gray-





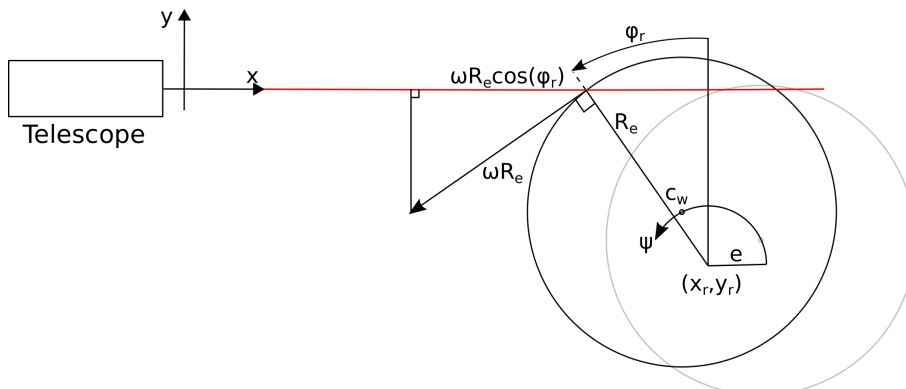

**Figure 6.** Schematic drawing of the influence of the wheel not rotating around its centre point. The wheel rotates around the point $(x_r, y_r)$ which is off-sat from the wheel centre, $c_w$, by $e$.

shaded circle, such that $\phi_r$ becomes negative, but because cosine is an even function the measured speed will still be the same. However, it is important to note that the measurement is not unaffected by the eccentricity because the radius of the wheel effectively becomes $R + e$ and is therefore larger than in the non-eccentric case. Also for very small tilt angles there will be a part of the wheel not being illuminated during a rotation and no measurement made. Another thing to notice is that this

conclusion will not hold for the thick beam, but as we have seen the narrow beam is really a very good approximation to the general case for angles larger than $\theta_1$ which are the angles of interest for the calibration.

### 3.6   Calibration procedure

As we have seen above our model predicts that there is a linear relationship between the ratio $\frac{V_{\text{LOS}}}{V_{\text{wheel}}}$ and the tilt angle $\theta$. This means that we can in principle measure the projected speed at any tilt angle larger than $\theta_1$ and extrapolate back to the speed at

$\theta_0$ ($\theta = 0$), i.e. where the bottom of the beam first touches the wheel, via Eq. (6). However, instead of a single measurement we choose to measure the projected speed over a range of tilt angles and fit a straight line to the measured values and in that way do the extrapolation based on a number of measurements. This is in practice done by slowly turning the tilt adjustment screw on the telescope while synchronously logging $V_{\text{wheel}}$, $V_{\text{LOS}}$ and $\theta$. This method furthermore has the advantage of not relying on a single lidar measurement which can be prone to discretisation uncertainty on the speed estimation. The change in angle cause

changes in the LOS speed that span several frequency bin widths and fitting over this range of angles will tend to average out the errors on the individual measurements.

     The difficulty with the method lies in establishing the angles $\theta_0$ and $\theta_1$ i.e. where the beam just starts to touch the wheel and when the entire beam is on the wheel, as illustrated in Fig. 7(a). In our setup the telescope is not perfectly aligned horizontally with the top of the wheel and therefore the laser beam is not perfectly horizontal at $\theta_0$ as shown i the figure and it is therefore

necessary to establish $\theta_0$ in a different way than from a direct angle measurement. For extremely shallow tilts the lidar only occasionally detects a signal, maybe due the slight eccentricity of the wheel or differences in the surface characteristic meaning





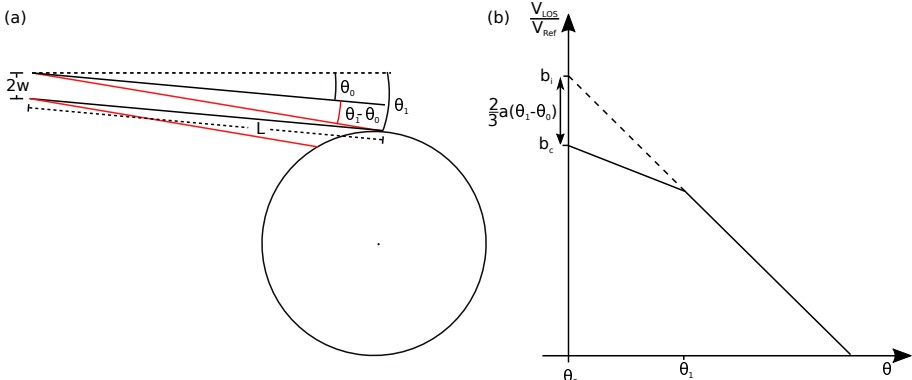

**Figure 7.** (a) Illustration of the definition of $\theta_0$ and $\theta_1$. $\theta_0$ is the angle compared to horizontal where the bottom of the beam first touches the wheel (beam drawn in black) and correspondingly $\theta_1$ is the angle where top of the beam first touches the wheel (red beam). (b) Illustration of the relation between the angles $\theta_0$ and $\theta_1$, and the fit intercept $b_i$ and the corrected intercept $b_c$.

that some parts of the wheel perimeter extends farther into the laser beam or reflects stronger than others. We choose the angle of this first sporadic signal as our best estimate for $\theta_0$.

The second angle, $\theta_1$, is more difficult to find and is more important for the overall calibration uncertainty. As the beam is slowly lowered from $\theta_0$ the gaps between meaningful measurements become shorter and fewer until eventually a continuous
lidar signal is achieved. This means that enough of the beam is now touching the wheel for a signal to be detected for all rotation angles and we choose the angle where this first occurs as the best estimate for $\theta_1$. Effectively we have thereby also estimated the beam radius as

$$w_{\text{est}} = \frac{L \cdot \tan \Delta \theta}{2},$$ (19)

where $\Delta \theta = \theta_1 - \theta_0$.

Another complication to the calibration arise due to the offset introduced by the finite beam width as explained in Sect. 3.4 and illustrated in Fig. 7(b). From the figure it is obvious that extrapolating from angles larger than $\theta_1$ will lead to an overestimation of the speed at $\theta_0$ and it is therefore necessary to compensate for this. We will do this via Eqs. (6) and (12) which states that the speed ratio is

$$\frac{V_{\text{LOS}}}{V_{\text{Wheel}}} = \begin{cases} 1 - \frac{1}{3} a\theta, & \text{for } \theta \leq \theta_1 \\ 1 - a\theta, & \text{for } \theta \geq \theta_1, \end{cases}$$ (20)

where $a = \frac{-L}{R}$ is the slope predicted by the models but instead we will use the slope of the actual linear regression while assuming that the $\frac{1}{3}$ relationship still holds. From this the overestimation can be found to be

$$\text{OE} = \frac{2}{3} a(\theta_1 - \theta_0).$$ (21)





In the end we therefore end up with an estimate of the ratio between speed measured by the lidar and the reference wheel speed given as

$$\left(\frac{V_{\text{LOS}}}{V_{\text{Wheel}}}\right)_{\text{est}} = b_i - \text{OE} = b_c, \tag{22}$$

where $b_i$ is the intercept of the fitted straight line at $\theta_0$ and $b_c$ is the compensated intercept. As we saw in Sect. 3.4, Eq. (20) is

strictly not correct because of the non-linear drop in $\frac{V_{\text{LOS}}}{V_{\text{Ref}}}$ close to $\theta_1$ as shown in Fig. 5, but for small values of $w$ the resulting error is small.

## 4  Uncertainties

In this section wee will give an estimate of the uncertainties associated with the various parameters going into the calibration and of course the overall calibration uncertainty. First the uncertainty on the reference speed is estimated, then the uncertainty

on the tangential speed measured by the lidar and finally we combine it into a total calibration uncertainty.

### 4.1  Uncertainty of reference speed (wheel)

From Eq. (1) we know that the speed of the wheel is given as

$$V_{\text{wheel}} = \omega R, \tag{23}$$

and the uncertainty on this can be obtained by applying the GUM model, Joint Committee for Guides in Metrology (2008) and

assuming that the uncertainties on the radius and on the rotational speed are uncorrelated

$$u_{V_{\text{wheel}}}^2 = \left(u_R \frac{\partial V_{\text{wheel}}}{\partial R}\right)^2 + \left(u_\omega \frac{\partial V_{\text{wheel}}}{\partial \omega}\right)^2 = u_R^2 \omega^2 + u_\omega^2 R^2, \tag{24}$$

which in relative terms becomes

$$\frac{u_{V_{\text{wheel}}}^2}{V_{\text{wheel}}^2} = \frac{u_R^2}{R^2} + \frac{u_\omega^2}{\omega^2}. \tag{25}$$

To get an estimate of the relative uncertainty we will assume an accuracy of $0.05$ mm for the wheel radius and that the

rotational frequency measurement is derived from a reference frequency that itself has an accuracy of $10^{-5}$ (10 ppm). Inserting into Eq. (25) gives

$$\frac{u_{V_{\text{wheel}}}^2}{V_{\text{wheel}}^2} = \left(\frac{0.05 \text{ mm}}{286.76 \text{ mm}}\right)^2 + \left(10^{-5}\right)^2 = \left(1.75 \cdot 10^{-4}\right)^2. \tag{26}$$

The standard uncertainty of the wheel speed is thus of the order of $0.02\%$.

The calibration flywheel is made of stainless steel which has a thermal expansion of the order $16 \cdot 10^{-6}$ /K, Cverna (2002).

Thus a change in temperature between the room where the radius was measured and the calibration room of 1 K will lead to a change in the reference speed of the same proportion. The temperature has not been monitored during these measurements but assigning an uncertainty of 3 K will lead to a relative uncertainty of $0.0048\%$ and therefore not contribute significantly to the overall uncertainty.





## 4.2 Uncertainty of $V_{\text{wheel}}$ measured by the lidar

With the calibration procedure suggested here the laser beam is slowly tilted more and more covering a wide range of projected speeds. Since the response in $V_{LOS}$ to a change in $\theta$ is almost linear it is possible to extrapolate back to the angle $\theta_0$ by fitting a straight line to the measured data. However, this indirect way of determining the tangential wheel speed is of course associated with different uncertainty contributions. Firstly the are the tilt angles, both the direct angle measurement but also very importantly the estimation of $\theta_0$ and $\theta_1$. Secondly there are uncertainties associated with the fit where noise in the measured values will lead to uncertainties in the forecasted slope and intercept.

Let us begin by looking at the uncertainty of the non-compensated intercept $b_i$. This is obtained from a linear regression which has an inherent uncertainty depending on the number of points in the regression and the level of noise in the measurements. The standard error of the regression, $SE$, can be calculated as

$$SE = \sqrt{\frac{n-1}{n-2}\left(\sigma^2_{\frac{V_{LOS}}{V_{\text{wheel}}}} - a^2\sigma^2_\theta\right)}, \tag{27}$$

where $n$ is the number of observations and $\sigma$ is the standard deviation, Lee and Seber (2003). $SE$ can be regarded as the standard deviation of the noise in the data and can be used to calculate the standard error of the estimated slope, $SE_a$ and intercept, $SE_b$

$$SE_a = \frac{SE}{\sqrt{n}\sigma_\theta}, \tag{28}$$

$$SE_b = \frac{SE}{\sqrt{n}}\cdot\sqrt{1+\frac{\langle\theta\rangle^2}{\sigma^2_\theta}}, \tag{29}$$

where $\langle\theta\rangle$ is the average of the measured tilt angles. As can be seen $SE_a$ and $SE_b$ depend inversely on the square root of number of observations and for the actual calibrations in this study the relative $SE_a$ is of the order $1\cdot10^{-4}$ or $0.01\%$ and $SE_b$ about a tenth of that which is so small in comparison to other contributions that they can be disregarded. Another contribution to the $b_i$ uncertainty is the estimation of $\theta_0$. Since the intercept between the extrapolation and the ordinate is taken to be the tangential wheel speed measured by the lidar it is clear that the position of $\theta_0$ and the uncertainty of this is of great importance for the measured speed and it's uncertainty. As mentioned in Sect. 3.6, $\theta_0$ is defined as the angle where the lidar first starts to pick up sporadic backscatter signals from the wheel surface but there is still an uncertainty associated with this due to the finite resolution, $\delta\theta$, of the inclinometer used to measure tilt angles. Assuming that the true $\theta_0$ is equally likely anywhere within the resolution range (a rectangular probability distribution), the uncertainty on $\theta_0$ can be found as

$$u_{\theta_0} = \frac{\delta\theta}{2\sqrt{3}}. \tag{30}$$

The squared uncertainty on the intercept due to $\theta_0$ is thus

$$u^2_{b_i} = \left(\frac{\delta\theta}{2\sqrt{3}}\cdot a\right)^2, \tag{31}$$





where $a$ is the slope of the extrapolation. The resolution of the tilt measurement is $\delta\theta = 0.01°$ leading to a standard uncertainty of $u_{b_i} = 0.028\%$ when applying a slope of $9.5\%/\circ$ as is found in the measurements, see Sect. 5.1. This uncertainty is of the same order as that of the wheel speed.

We know that $b_i$ leads to an overestimation of the speed measured by the lidar and that we need to compensate for this. From
Eq. (22) we know that the calibration value compensated for the beam width is given as

$$b_c = b_i - \frac{2}{3}a\Delta\theta, \tag{32}$$

and the uncertainty on this can thus be estimated through

$$
\begin{aligned}
u_{b_c}^2 &= & u_{b_i}^2\left(\frac{\partial b_c}{\partial b_i}\right)^2 + u_a^2\left(\frac{\partial b_c}{\partial a}\right)^2 + u_{\Delta\theta}^2\left(\frac{\partial b_c}{\partial \Delta\theta}\right)^2 \\
&= & u_{b_i}^2 + u_a^2\left(-\frac{2}{3}\Delta\theta\right)^2 + u_{\Delta\theta}^2\left(-\frac{2}{3}a\right)^2,
\end{aligned}
\tag{33}
$$

where we have assumed that the uncertainties of the input parameters are uncorrelated. $u_{b_i}$ and $u_a$ have both been estimated above and in the following we will find an estimate for $u_{\Delta\theta}$.

We will assume that the measurement of $\theta_0$ and $\theta_1$ are each associated with two uncertainty contributions $u_{\theta_M}$ and $u_{\theta,D}$. $u_{\theta,M}$ is related to the absolute measurement of $\theta_0$ or $\theta_1$ and could e.g. be due to a gain or offset error, and $u_{\theta_D}$ is a discrimination uncertainty due to the finite resolution of the inclinometer. $\theta_{0M}$ and $\theta_{1M}$ are correlated because they are measured using the
same gauge whereas $\theta_{0D}$ and $\theta_{1D}$ are uncorrelated. We can therefore find the squared uncertainty on $\Delta\theta = \theta_1 - \theta_0$ as

$$
\begin{aligned}
u_{\Delta\theta}^2 &= & u_{\theta_{0M}}^2\left(\frac{\partial \Delta\theta}{\partial \theta_0}\right)^2 + u_{\theta_{0D}}^2\left(\frac{\partial \Delta\theta}{\partial \theta_0}\right)^2 + u_{\theta_{1M}}^2\left(\frac{\partial \Delta\theta}{\partial \theta_1}\right)^2 + u_{\theta_{1D}}^2\left(\frac{\partial \Delta\theta}{\partial \theta_1}\right)^2 + 2u_{\theta_{0M}}^2\frac{\partial \Delta\theta}{\partial \theta_0}\frac{\partial \Delta\theta}{\partial \theta_1}u_{\theta_{0M}}u_{\theta_{1M}}r \\
&= & u_{\theta_{0M}}^2 + u_{\theta_{0D}}^2 + u_{\theta_{1M}}^2 + u_{\theta_{1D}}^2 - 2u_{\theta_{0M}}u_{\theta_{1M}}r, \quad (34)
\end{aligned}
$$

where $r$ is the correlation coefficient. If assume that $u_{\theta_{0M}} = u_{\theta_{1M}} = u_{\theta_M}$ fully correlated and $u_{\theta_{0D}} = u_{\theta_{1D}} = u_{\theta_D}$ are uncorrelated we end up with

$$u_{\Delta\theta}^2 = 2u_{\theta_M}^2 + 2u_{\theta_D}^2 - 2u_{\theta_M}^2 = 2u_{\theta_D}^2, \tag{35}$$

where $u_{\theta_D}$ is the same as $u_{\theta_0}$ in Eq. (30).

Now, $\Delta\theta$ is in essence our best estimate of the beam width as expressed through Eq. (19) but the validity of this assumption is associated with some uncertainty. As we shall see in Sect. 5.1 the beam radius estimated with this method does resemble what we would expect from a theoretical calculation of the beam radius, but on the other hand there is no reason to believe it
to be a completely correct estimate either. This uncertainty must be incorporated into $u_{\Delta\theta}$ and we do this by adding the term $u_{\theta_w}$ such that

$$u_{\Delta\theta}^2 = 2u_{\theta_D}^2 + u_{\theta_w}^2. \tag{36}$$

As mentioned above $u_{\theta_w}$ is quite large and in the following we will assume it to be $100\%$ in relative terms such that $u_{\theta_w} = \Delta\theta$. For the smallest beam tested in this study we have $\Delta\theta = 0.01°$ and this leads to an uncertainty originating from
$u_{\Delta\theta}$ of about $0.069\%$ out of a total of $0.074\%$ and $u_{\Delta\theta}$ is thus seen to be the dominant term in $u_{b_c}$.



### 4.3 Overall calibration uncertainty

We can finally find the overall measurement uncertainty. The lidar estimate of the wheel speed is the compensated calibration constant times the reference wheel speed

$$V_{\text{LOS}} = V_{\text{wheel}} \cdot b_c, \tag{37}$$

and the squared uncertainty therefore becomes

$$u_{V_{\text{LOS}}}^2 = u_{V_{\text{wheel}}}^2 b_c^2 + u_{c_c}^2 V_{\text{wheel}}^2, \tag{38}$$

and relative to $V_{\text{wheel}}$

$$\left( \frac{u_{V_{\text{wheel}}}}{V_{\text{LOS}}} \right)^2 = \left( \frac{u_{V_{\text{wheel}}}}{V_{\text{wheel}}} b_c \right)^2 + u_{b_c}^2. \tag{39}$$

This means that $u_{b_c}$ and therefore $u_{\Delta\theta}$ is the main contributor to the overall calibration uncertainty.

## 5   Calibration measurements

In this section will be presented calibration measurements made with different beam widths and for different reference speeds.

### 5.1   Beam width

It is clear from the uncertainty analysis in Sect. 4 that the beam radius, expressed through $\Delta\theta$, is of great importance for the overall calibration uncertainty. However, how to establish the beam width is not trivial even though our beam is well-behaved

and can be well approximated by a pure Gaussian beam and the equations describing this, see Sect. A. For Gaussian beams in general the $1/e$ or $1/e^2$ width of either the electrical field or irradiance is often used to define the beam radius but in our case it must be defined as the width from which it is possible to detect a signal and this depends on several parameters such as the detection threshold, scattering properties of the wheel surface, and also the angle between beam and wheel surface. The best way we have of quantifying this is therefore to measure the tilt angles where we first detect a signal and where we constantly

see a signal, respectively. It is clear that there is no guarantee that these angles represent the beam width and it is therefore associated with a significant uncertainty but it is the best estimate we can make with the data at hand.

Following the procedure outlined in Sect. 3.6 we have carried out a series of calibrations with different focus of the beam ranging from about 1 m to 5 m resulting in different beam widths at the wheel. Fig. 8 shows the measured beam radii as function of focus distance calculated from Eq. 19. Shown is also the theoretical $1/e^2$ radii of the irradiance calculated using

the standard equations for Gaussian beams. The first thing to notice is that the shape of the two curves resemble each other quite well indicating that this way of estimating the width does actually capture some of the truth about it at least relatively. The values of the calculated widths are about three times higher than the measured, but this is not too disturbing since we are not expecting the measured width to represent the $1/e^2$-width but rather the width from where we can detect a signal. More



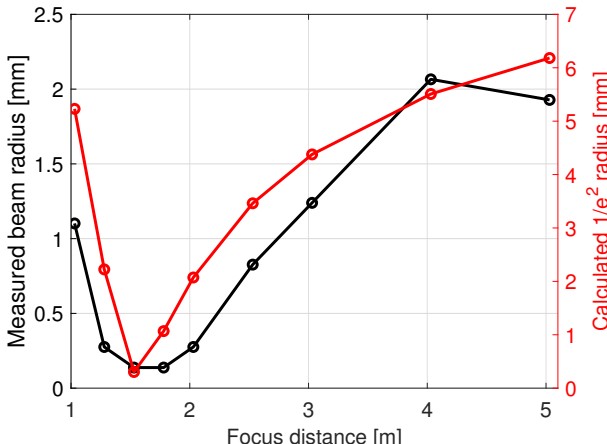

**Figure 8.** Comparison between the measured beam widths and the theoretically calculated $1/e^2$ radius as function of focus distance.

concerning is that the minimum around $1.5$ m is not nearly as sharply defined for the measured values as for the theoretical and some of this could possibly be due the finite resolution of the angle measurement but probably not all of it. The minimum beam radius of $0.14$ m located at $1.53$ m and $1.78$ m actually corresponds to $\Delta\theta = 0.01°$ which is the same as the angle measurement resolution. Finally, there is the point at $4.03$ m which could look like an outlier; the beam width should not be higher with the

focus at $4.03$ m than at $5.03$ m but this has not been clarified. All in all it is very difficult do determine the beam radius and it is thus associated with a large uncertainty. In order to put some numbers on we estimate a relative uncertainty of $30\%$ for the larger beam radii and $50\%$ and even $100\%$ for the smallest beams. The resulting absolute uncertainties can be seen in Table 2 together with the theoretical and measured beam width for each focus distance.

Figures 9 and 10 show two examples of the calibrations made. Figure 9 is made with the laser beam focused at $1.53$ m and

thus with the beam waist located very near the top of the wheel so that the beam width on the wheel is about as small as the setup allows while in Fig. 10 the focus is placed at $2.53$ m. The mean reference speed is $10.93$ m/s. It is clearly seen how $\frac{V_{\text{LOS}}}{V_{\text{wheel}}}$ in general falls off linearly as function of tilt angle as predicted by the models. However, it is also seen that on top of this trend are some discrete steps, something that is also reflected in the residual plot. These steps are due to the narrow beam width resulting in the range of sensed speed being smaller than the resolution of the lidar's Doppler spectrum resulting in the

speed estimation to jump from bin to bin quite abruptly. In contrast, this feature is almost completely gone in Figure 10 where the beam is much bigger and therefore a larger range of radial velocities covered in each measurement widening the Doppler signal over more bins. This binning effect has an impact on the fit result through the standard error on the slope and intercept which is indeed higher for the narrow beam, but as mentioned in Sect. 4.2 this effect is still much smaller than other uncertainty contributions. This smoothing effect of the regression can also be seen in the residual plots (lower panel) which have average

values of essentially 0 (ranges between $2.3 \cdot 10^{-14}$ and $-2.1 \cdot 10^{-14}$ for Fig. 9 and Fig. 10, respectively) meaning the fit is very close to the average. The red line in the top panel of the figures is the least-squares fit of a straight line to the measurement





**Table 2.** Theoretical and measured beam widths together with estimated uncertainties for the different focus distances used.

| Focus distance | Theo. beam radius | Meas. beam radius | Beam radius uncertainty |
|:---:|:---:|:---:|:---:|
| [m] | [mm] | [mm] | [mm] |
| 1.03 | 5.23 | 1.10 | 0.41 |
| 1.28 | 2.22 | 0.28 | 0.14 |
| 1.53 | 0.30 | 0.14 | 0.14 |
| 1.78 | 1.07 | 0.14 | 0.14 |
| 2.03 | 2.07 | 0.28 | 0.21 |
| 2.53 | 3.46 | 0.83 | 0.28 |
| 3.03 | 4.38 | 1.24 | 0.41 |
| 4.03 | 5.51 | 2.07 | 0.69 |
| 5.03 | 6.19 | 1.93 | 0.69 |

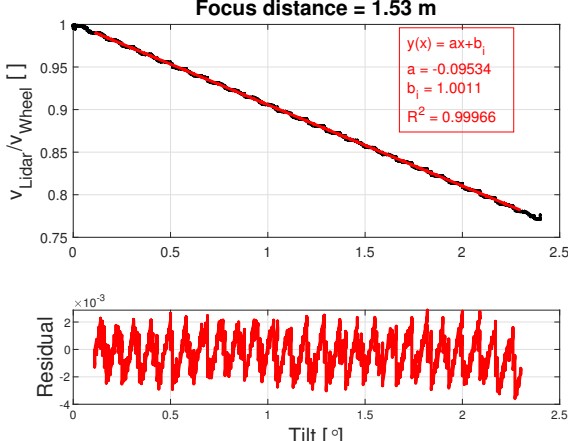

**Figure 9.** Example of calibration measurement made with a focus setting of $1.53$ m meaning that the waist of the beam is placed very near the top of the wheel. The black curve is the measurement data and the red a least-squares fit of a straight line to the data. In the lower panels is shown the residuals of the fit.

data over the range of tilt angles indicated by the extend of the red line itself. The fit ranges over angles from $\theta_0 + 0.1°$ to $0.1°$ before the maximum measured tilt angle. The resulting fit parameters, slope and intercept, are shown in the insets.

Figure 11 shows the fit slopes and intercepts for all nine calibrations as function of focus distance. According to the narrow beam model the slope of $\frac{V_{\text{LOS}}}{V_{\text{wheel}}}$ is $-\frac{L}{R}$ which with the parameters specified in Table 1 equals $-9.60\,\%$ per degree and from the
5    figures it is seen that the measured slopes range from $-9.496\,\%/\circ$ to $-9.537\,\%/\circ$. This is not a large difference but it is still



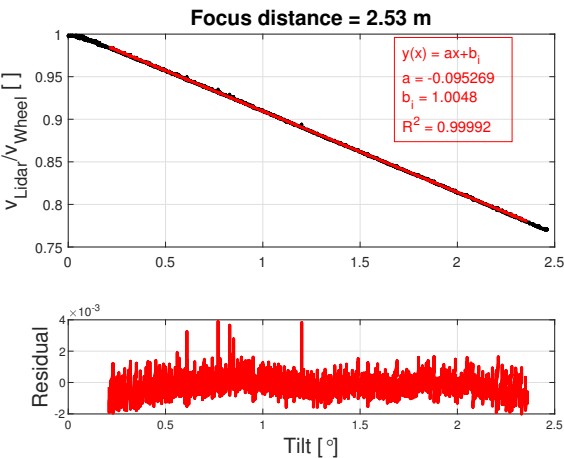

**Figure 10.** Example of calibration measurement made with a focus setting of $2.53$ m. The black curve is the measurement data and the red a least-squares fit of a straight line to the data. In the lower panels is shown the residuals of the fit.

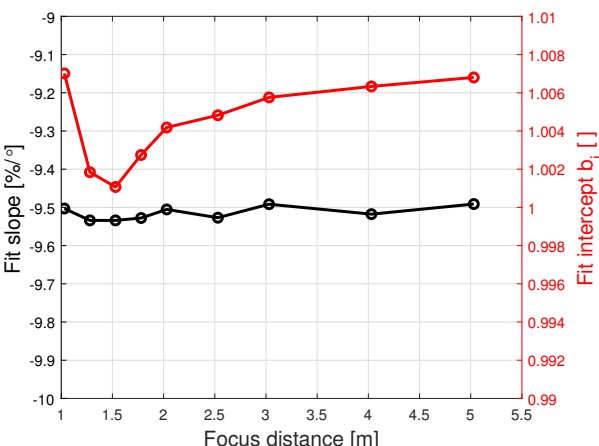

**Figure 11.** Fit slope and intercept as function of focus distance. The intercept values show a clear minimum at $1.53$ m where the beam with at the wheel is smallest clearly illustrating the need for compensating these results.

larger than the estimated uncertainty and it is more or less the same for all calibrations and not just one or two outliers. The reason behind this deviation is not known.

More interesting for the calibration purpose is of course the fit intercept which is shown in red in Fig. 11. It is clearly seen that the intercept overestimates as expected and in shape curve looks a lot like the measured beam width in Fig. 8. This highlights the need to compensate the calibration result for this. This has been done in Fig.12 where the red curve shows the compensated intercept values and the black curves represent the estimated standard uncertainties. The compensation has clearly brought the



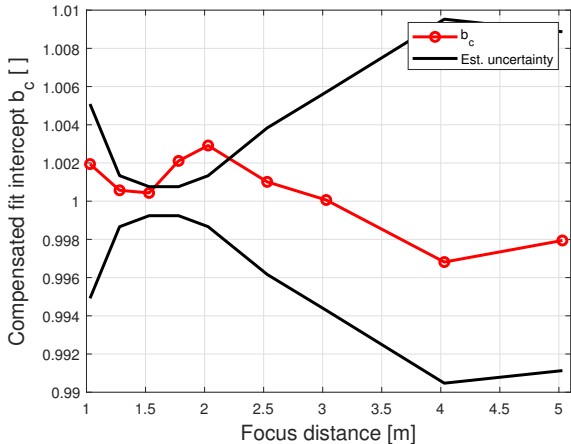

**Figure 12.** Compensated regression intercepts together with the estimated standard uncertainties.

intercept closer to $1$ with the maximum and minimum placed approximately $0.3\%$ on either side. Most of the points are within or very close to the estimated standard uncertainty with the exception of the points at $1.78\,\mathrm{m}$ and $2.03\,\mathrm{m}$. This is probably due to the measurement of $\Delta\theta$ which seems low compared to the theoretical value as seen in Fig. 8 and therefore the compensation becomes too weak. The combined uncertainties calculated using the equations derived in Sec.4 range from about $0.08\%$ for

the narrow beams up to about $0.9\%$ for the widest beams, and it is clearly seen how the shape of the uncertainty curve follows that of the measured beam width in Fig. 8. This is due to the term $u_{\theta_w}$ which we have estimated to be equal to the value of $\Delta\theta$ and which is dominating. This underlines the importance of a good estimate of the beam width.

## 5.2 Different reference speeds

In this section we present the results of calibrations made with different reference speeds but a fixed focus distance of $1.53\,\mathrm{m}$,

i.e. with the smallest possible beam width at the wheel.

The tested reference speeds range from about $3.3\,\mathrm{m/s}$ to $17.3\,\mathrm{m/s}$, and Figs. 13 and 14 two examples of measurements and fits made at $5.44\,\mathrm{m/s}$ and $13.89\,\mathrm{m/s}$, respectively. In both cases there is a very good agreement with what is expected from the model as well as with the results in Sect. 5.1. It is noted that the characteristic staircase shape also seen in Fig. 9 is very pronounced in both these figures, but that the length of each "step" seems to change with the reference wheel speed. This is

because that what is plotted is the ratio between measured and reference speed and for low reference speeds the bins of the Doppler spectra becomes relatively larger as function of tilt angle.

Fig. 15 shows the resulting fit parameters for all the tested reference speeds. We see that the slope of the fit lies between $-9.51\%/°$ $-9.56\%/\circ$ and is more or less constant across the tested speeds. Also the fit intercept is very close to constant for the first five tested speeds and is bounded within $0.998$ and $1.002$, and thus within the estimated uncertainty, but the last point

at $17.26$ m/s stands out a bit. Here the intercept drops to below $1$, but is still within the uncertainty. A possible explanation





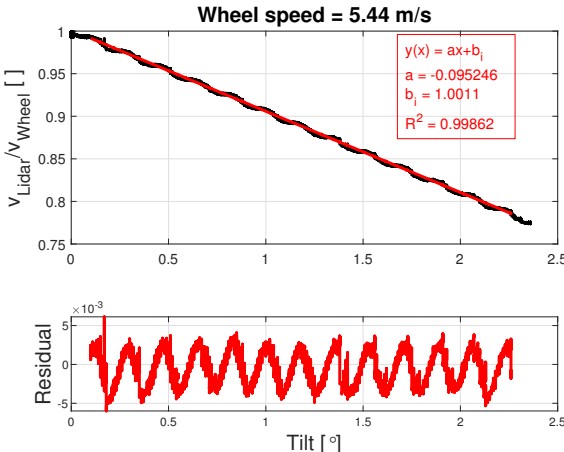

**Figure 13.** Example of calibration measurement made with a reference speed $5.44\,\mathrm{m/s}$ and a focus distance of $1.53\,\mathrm{m}$. The black curve is the measurement data and the red a least-squares fit of a straight line to the data. In the lower panels is shown the residuals of the fit showing some distinct oscillations because of the speed estimation jumping from bin to bin due to the very narrow beam.

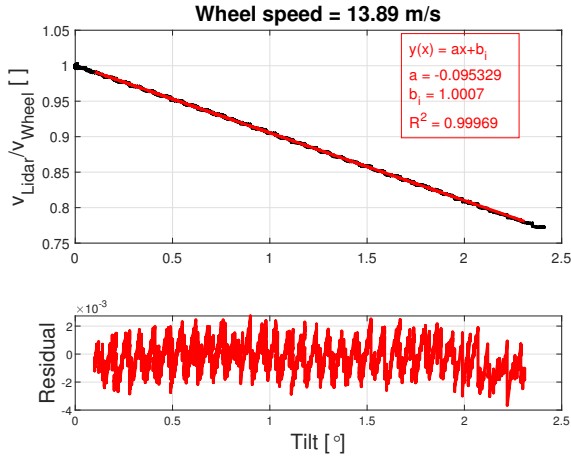

**Figure 14.** Example of calibration measurement made with a reference speed $13.89\,\mathrm{m/s}$ and a focus distance of $1.53\,\mathrm{m}$.

for this is that at 17 m/s the wheel rotates at around 9.5 revolutions per second which is quite fast and the entire rig including transceiver and inclinometer starts to shake which limits both lidar and angle measurement.

Again the regression intercepts have been compensated and the result is shown in Fig. 16 together with the uncertainties. Because of the small beam width used in these measurements the difference between $b_i$ and $b_c$ is very small. The assumed uncertainty on $\Delta\theta$ is the same as for the same focus distance in the previous section and the resulting combined standard uncertainty is about $0.08\%$ which most of the points lie close to.

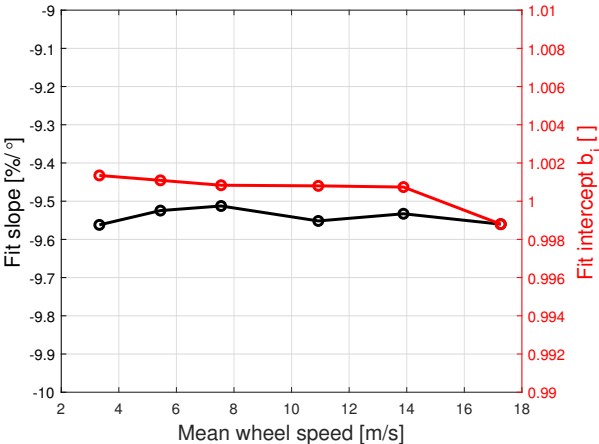

**Figure 15.** Results of fits for different reference speeds. Fit intercept not compensated for $w$.

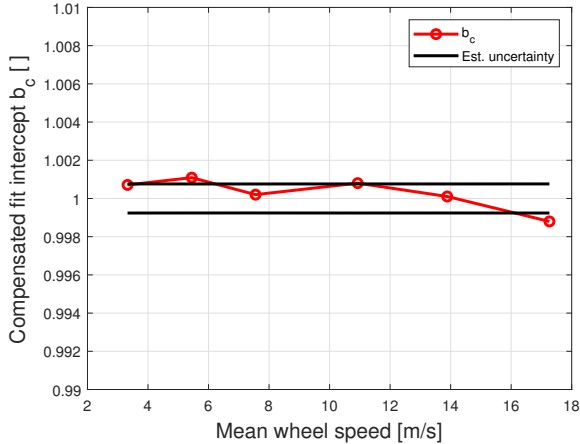

**Figure 16.** Compensated fit intercepts together with the estimated standard uncertainties.

## 6 Discussion

It can seem paradoxical to use a flywheel to calibrate a Doppler wind lidar when the parameter we want to measure, the peripheral speed, is the one thing the lidar can not measure. Instead it seems more straight forward to measure a linear motion along the direction of the beam and this might very well be the case. For example, besides directly measuring the desired parameter, the uncertainties introduced by the angle measurement and assessing the zero-point of the angle scale together with the beam width can be alleviated. However, there are also arguments for using the flywheel; as discussed earlier by scanning a range of speeds and fitting the inherent uncertainty due to discretisation is reduced, and the symmetrical nature of the wheel makes it easy to obtain a very stable reference speed whereas with a linear motion the target would probably have to be moved



back and forth and thus accelerated up to a known speed repeatedly. This would then require the position of the reference target to logged together with it's speed which again demands a more complicated geometrical model. Another idea could be to measure in a range of angles covering the direction toward the centre of the wheel. In this way, a zero-point defined as the angle where the beam is perpendicular to the wheel surface could be established as where no speed is measured thus alleviating

the problems with finding $\theta_0$, and possibly $\theta_1$, seen above. Unfortunately, the calibration setup in it's present state does not allow for such a measurement to be made due to limitations in the attainable tilt angles.

## 7    Conclusions

Inspired by a similar concept commonly used for calibrating LDAs we have constructed a setup for calibrating coherent Doppler wind lidars based on a spinning flywheel with the lidar beam skimming the wheel periphery. The setup is made in

such a way that the laser beam can be tilted and thus probing different projections of the wheel's tangential speed. A simple model shows that there is a linear relation between the beam tilt angle and the measured LOS speed and this can be utilised to extrapolate back to the true tangential speed at zero tilt; the one angle otherwise impossible to measure at because the physical overlap between wheel surface and laser beam disappears. The model takes into account the finite width of the laser beam but only under the assumption that the beam is collimated while in reality the beam used in the tests is actually focused in order

to control the beam radius. The model also forms the basis of the uncertainty analysis which concludes that a total calibration standard uncertainty of about $0.1\%$ can be achieved with this setup. The uncertainty analysis reveals that the main contributor to the total uncertainty is the finite radius of the laser beam and in order to reduce the uncertainty it is essential to determine this better than we have been able to achieve so far. Calibration measurements performed at different reference speeds and with different beam widths all show a good agreement with the model and confirms that the lowest uncertainty is achieved when the

beam width is minimised.

*Code and data availability.* The measurements and scripts for data analysis is available via Pedersen (2020).

## Appendix A: Width of Gaussian beam

The theoretical beam width at the top of the wheel can calculated by appropriate combination of the following two equations: The width of an untruncated Gaussian beam at a distance $x'$ from the beam waist can be calculated through

$$w(x') = w_0\sqrt{1 + \left(\frac{\lambda x'}{\pi w_0^2}\right)^2},$$ \hfill (A1)



where $w_0$ is the width at the waist and $\lambda$ is the laser wavelength, Siegman (1986). Similarly can $w_0$ with the waist placed a distance $x$ from the focusing lens be found as

$$w_0(x) = \sqrt{\frac{w_l^2 - \sqrt{w_l^4 - 4\left(\frac{\lambda x}{\pi}\right)^2}}{2}}, \tag{A2}$$

where $w_l$ is the beam width at the lens.

5  *Author contributions.* Anders Tegtmeier Pedersen: Measurements, data processing, data analysis, model development, manuscript writing. Michael Courtney: Conceptual idea, model development, manuscript writing.

*Competing interests.* The authors declare that they have no conflict of interest.

*Disclaimer.* TEXT

*Acknowledgements.* This work has been performed under the Danish EUDP project *TrueWind*, journal number 64015-0635.

10  Professor Torben Mikkelsen and Steen Andreasen are gratefully acknowledged for initiating, designing, and constructing the calibration rig. All members of the DTU Wind Energy, TEM, Measurement Systems Engineering team are thanked for their great help and support.



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
