# Peer review of "Flywheel calibration of a continuous-wave coherent Doppler wind lidar"

_Atmospheric Measurement Techniques, 2020_

## Referee Comment (RC1) · Anonymous Referee #1 · 11 Aug 2020

Pedersen and Courtney present a study dealing with a calibration setup for Doppler wind lidars. The setup is a flywheel that is connected to the lidar that is to be calibrated with a solid frame and a tilting mechanism for the lidar. The authors do a great job in deriving the math to obtain a measurement of the flywheel's rotational speed from a sweep of lidar measurements across the wheel's curved surface. They also provide a detailed derivation of the uncertainty components which is best practice. The manuscript is however very vague when it comes to the actual potential of the method for lidar calibration, this is mostly because no targeted accuracy of the calibration is defined to which the results are compared. This makes it difficult to assess the relevance of this work. I recommend major revisions before the manuscript can be accepted for

AMT. Below, I give some general coments and more specific comments to parts of the manuscript.

**0.1 General comments**

- The style of the text is very pessimistic and suggests that the authors do not believe in the value of their work. Why would they want to publish it in this case? I personally believe that there is a lot of value in this work, and I suggest that the authors try to reformulate in a more objective, quantitive way.

- The mathematical formulations should be revisited. Some of the equations are hard to read. Double subscripts should be omitted, variable names should be consistent and using $u$ for uncertainties in the context of wind measurements is a bit unfortunate. A nomenclature in the appendix would be helpful.

- The manuscript is a bit hard to read, because the structure is unclear from the beginning, and the goal of the work is not well described in the beginning. Some of the derivations of rather simple geometric relations could be shortened or moved to the appendix to make the manuscript text flow more nicely.

- My main criticism for the content of the manuscript is, that it does not describe well what are the requirements for calibration accuracy in practice. The authors only briefly mention possible sources of error in Doppler wind lidars, but do not give an idea about the quantity of such errors. How accurate do lidar measurements have to be for the industry or for research purposes? Maybe the calibration accuracy here is good enough after all? In fact, it cannot be known from this experiment if differences between wheel speed estimation through the calibration and reference wheel speed are due to inaccuracy in the calibration setup or the lidar itself.

- Until the last sentence of the discussion Section 6 I was wondering if it was not a possibility to measure from zero wind speed perpendicular to the wheel surface to get a more continuous calibration curve. It is a bit unsatisfactory and hard to understand why the tilt angle of the lidar cannot be mechanically adjusted for that.

**0.2 Specific comments**

- Title: The title is a little bit misleading, because I was expecting to see results of the calibration of the lidar, but actually there are only evaluations about the accuracy of the calibration itself, not the lidar.

- Introduction:

  – p.1,l.20: I think it would be worthwhile to expand on the kind of laser errors that can cause errors of line-of-sight velocity to better motivate the work.
  – p.2, l.1: Shinder et al. 2013 should be inside the parantheses. The citation style should be checked throughout the manuscript.

- Section 2:

  – p.2,l.22ff: Since this is a commercial product, I think it would be sufficient to name the manufacturer and give the accuracy of rotation speed measurement. Measurement principle is not so important for this study.
  – Table 1: I recommend to divide the table in two, separating calibration rig properties and lidar properties.

- Section 3:

  – All equations: I suggest to use $\varphi$ (varphi) to be consistent with the plots.
  – p.4, l.2: Just a suggestion for wording: I do not think it is a severe problem, just a technical one.

– p.4, l.7: Please expand why the resolution of the Doppler spectrum is a problem. It does not define the resolution of the velocity measurement, because the typical MLEs interpolate the spectrum to estimate the Doppler shift. I think the main benefit of several tilt angles / wheel speeds is that the tangential speed can then be extrapolated!?

– p.4, l.24: I would not call it a trick, it is geometry after all.

– p.6, l.8f: I think the connection between L'Hospital and the following Taylor's expansion is not made very clear. Maybe Eqs. 9-10 could even be put in the appendix and just be referenced at this point to make it easier to follow the main storyline of the paper.

– p.7, l.11: Why does part of the beam even have to go above the wheel? Wouldn't it make it easier to just choose tilt angles where the whole beam is on the wheel?

– p.9, Eq.17: I think the whole derivation does not have to appear here, especially since the 3D model does not have a significant benefit over the 2D model. It could still be put in the appendix.

– p.10, l.10f: Could the beam width maybe be estimated from the tilt angle sweep if it is done so far until none of the beam hits the wheel any more? Or if instead of trying to hit the tangential, the beam was set to hit the wheel frontal, which would lead to zero velocity measurement?

– p.12, l.11f: It is not clear to me why the extrapolation from angles larger $\theta_1$ is not the real tangential speed ratio ($b_i = 1$!?).

• Section 4:

– p.14, Eq.27: It would be good to introduce a variable for $\frac{V_{\text{LOS}}}{V_{\text{wheel}}}$ and use this as subtext in this equation.

• Section 5:

- p.16, l.26: "quite well", "some of the truth" and "at least relatively" are such weak statements that I strongly recommend to reformulate.
- p.20, l.18: $\%/\circ$ is not a good notation and it is also mixed with $\%/°$.
- p.23, l.2ff: I was wondering about this possibility throughout the whole derivation of the calibration models for $\theta < \theta_1$. Maybe it should be explained earlier, why it is not possible.
- p.23, l.25: "its" instead of "it's".
- p.23, l.19f: Is it enough to calibrate continuous wave lidars at a minimised beam width? What would an industrial calibration have to cover?
* * *

---

## Referee Comment (RC2) · Anonymous Referee #3 · 17 Sep 2020

**General comments**

The paper of Pedersen and Courtney is well structured and written. They are thorough in their way of deriving models, uncertainties and finally comparing it to measurements. Their approach to use a flywheel in combination with a Doppler wind lidar seems new and worth publishing (after minor revisions).

**Specific comments**

Concerning the estimate of $\theta_1$ (P12L5-6 and P16L18-21). Did you also measure backscatter? Why not look at the plot of tilt vs backscatter? I suppose it should be increasing from $\theta_0$ to $\theta_1$ strongly and afterwards only slightly (if at all)? If it turns out

that the increase in backscatter shows no point of change around $\theta_1$, just make a different set up where you place something (with a sharp edge) on top of the (not turning) wheel and tilt and measure the backscatter as the beam hits the object first partially and then completely. Now the plot should just show an increase in backscatter from the angle where the beam touches the object partially to the angle where it hits it completely and after that stay constant? The difference could be an estimate for $\Delta\theta$? (To avoid backscatter from walls, use window or something that reflects at an angle as background?). If my thoughts on this are correct but a new setup/measurement is too time consuming please address/discuss this appropriately in the document.

In section 3.1 you describe your approach to rotate around the transceiver lens *for the model*. But I did not see you describing where the tilt-axis lies relative to the telescope *for your measurements*. If you raise/lower just the end so it rotates around its lens (As the caption of Fig.2 suggests), please include this information somewhere. If the tilt axis is not going through the lens please also address this (maybe add some text that makes clear that the changes/differences to L and $y_r$ are negligible)

Although it is obvious that no wind speeds are measured because the title says "flywheel calibration" - It is after all a "Doppler wind lidar". As someone who uses a Doppler wind lidar to measure wind speeds it feels a bit weird to read the whole paper and end up just with a calibration "for rotating steel". I of course prefer a lidar that goes through such a quality check, but still... the journal is ATMOSPHERIC Measurement Techniques... Maybe you could add a paragraph about how and why this translates to wind measurements or state that this calibration is meant more as a necessity/possibility than as a sufficiency for Doppler wind lidars quality? (This may be a matter of taste... if you feel all is clear by using the word "calibration" that is also fine)

**Technical corrections**

Code and data availability: Pedersen, A. T.: Flywheel calibration of coherent Doppler wind lidar - data, https://doi.org/10.11583/DTU.11991189, 2020.

gives me -> Page not found

https://amt.copernicus.org/preprints/amt-2020-88/ says "Anders Tegtmeier Pedersen and Pedersen Courtney" and the paper itself says "Anders Tegtmeier Pedersen and Michael Courtney"

There are a bunch of "r"-index is missing: EQ9, P6L15-16, EQ10, P7L5/6/9/11

In the text and figures, you use different styles of the Greek phi ($\phi/\varphi$). Please make this consistent.

You use "best" 5 times in the document. When you say "our best" I get it, but just "best" is a bit bold. Maybe rephrase some occurrences.

P2L19-20 The reference to Fig.1 makes it seem like we should be able to see the inclinometer on top of the telescope. I don't see it... maybe use labels / zoom in the Fig.1 or if it is an old picture without inclinometer move the Fig.1 reference to one sentence earlier?

P6L8 "this" is ambiguous. Maybe use "the right hand side divided by $V_{wheel}$"

EQ8 $\theta$ should not be there, right?

P6L15-16 not listed as equations

P8L15 should be $y_r - R \le y \le y_r + R$ or $-R \le y - y_r \le R$, right?

EQ17 $\phi$ is missing "r"-index

EQ17 I don't follow the last equal sign. Please explain, expand or correct.

P10L4 "as long the" -> "as long as the"

P11L14 "cause" -> "causes"

P12L10 "arise" -> "arises"

P13L8 "wee" -> "we"

P14L5 "the are" -> "there are"

P14L19 "shown i the" -> "shown in the"

P14L18-20 Please check the sentence structure again. The last bit doesn't seem right. Maybe "way than" -> "way other than". Maybe even rephrase as "direct angle measurement" is ambiguous. Did you mean "direct angle measurement from inclinometer"?

P15L8 "assume" -> "assumed"

EQ38 index $c_c$ should be $b_c$

EQ39 I don't follow this transformation. Did you swap "wheel" and "LOS"?

P17L3 "0.14m" -> "0.14 mm"

P17L7 "Table2" -> "Table 2"

P17L15-17 Sentence looks wrong. Maybe "measurement widening" -> "measurement is widening"?

P19L4 "shape curve" -> "shape the curve"

**Cosmetic suggestions**

(These need not be address)

P5L4 "rearranging and inserting into Eq. (1)" with the page break - I found it hard to follow (first time reading), that you also use eq.3 in this step.

Fig.6 An arrow for $wR_e \cos(phi_r)$ could be nice, but I guess it overlaps with the red line? (Maybe dashed or dotted arrow?)

P13L1 double usage of "end" is hard to read

P15L22-26 I would rephrase it... explain/motivate $u_{\theta_w}$ differently.

P17L6 "relative uncertainty of" -> "relative uncertainty $u_{\theta_w}$ of"

P17L7 "absolute uncertainties can" -> "absolute uncertainties $u_{\Delta\theta}$ can"

Fig.9/10 different colour for residuals lines?

---

## Author Comment (AC1) · 15 Oct 2020

The authors gratefully thank the reviewer for a fair review and constructive criticism. Response to the individual comments is provided in the attached letter ('amt-2020-88-supplement.pdf')

On behalf of the authors, Anders T. Pedersen

Please also note the supplement to this comment:
https://amt.copernicus.org/preprints/amt-2020-88/amt-2020-88-AC1-supplement.pdf

---

## Author Response (AR1)

**Response to reviewer comments**

We gratefully thank the reviewers for their comments and criticism which we find both well-founded and constructive. Below we answer the comments individually in red and where appropriate also provide our suggested corrections to the manuscript.

On behalf of the authors

Anders T. Pedersen

**Reviewer #1**

Pedersen and Courtney present a study dealing with a calibration setup for Doppler wind lidars. The setup is a flywheel that is connected to the lidar that is to be calibrated with a solid frame and a tilting mechanism for the lidar. The authors do a great job in deriving the math to obtain a measurement of the flywheel's rotational speed from a sweep of lidar measurements across the wheel's curved surface. They also provide a detailed derivation of the uncertainty components which is best practice. The manuscript is however very vague when it comes to the actual potential of the method for lidar calibration, this is mostly because no targeted accuracy of the calibration is defined to which the results are compared. This makes it difficult to assess the relevance of this work. I recommend major revisions before the manuscript can be accepted for AMT. Below, I give some general comments and more specific comments to parts of the manuscript.

0.1 General comments

• The style of the text is very pessimistic and suggests that the authors do not believe in the value of their work. Why would they want to publish it in this case? I personally believe that there is a lot of value in this work, and I suggest that the authors try to reformulate in a more objective, quantitive way.

We feel that it is our job to be critical about own work and point toward the weak points. However, there should be no doubt that we believe in the value of the work and have therefore reformulated the discussion in a more objective and positive way.

'Nevertheless, the presented measurements and analysis show that the proposed calibration method is not only practically feasible but could actually lead to a significant reduction in calibration uncertainty compared to the current practice. However, there could also be other methods for achieving a similar calibration result. For instance, it seems more straight forward to measure a linear motion along the direction of the beam and this might very well be the case because besides directly measuring the desired parameter, the uncertainties introduced by the angle measurement and assessing the zero-point of the angle scale together with the beam width can be alleviated.'

Also, we have in the introduction tried to clarify the objective of the work and in that way show that the goal is reached, see later comment.

• The mathematical formulations should be revisited. Some of the equations are hard to read. Double subscripts should be omitted, variable names should be consistent and using u for uncertainties in the context of wind measurements is a bit unfortunate. A nomenclature in the appendix would be helpful. Mathematical expressions have been revisited and corrected; some derivations moved to appendices, see also Specific comments.

The symbol used for uncertainty has been changed to (capital) U.

We prefer keeping the double subscript symbols as we would otherwise need to introduce new variables which would quickly become very confusing.

• The manuscript is a bit hard to read, because the structure is unclear from the beginning, and the goal of the work is not well described in the beginning. Some of the derivations of rather simple geometric relations could be shortened or moved to the appendix to make the manuscript text flow more nicely.

Some less important derivations have been put in appendices and the objective of the manuscript clarified in the introduction.

• My main criticism for the content of the manuscript is, that it does not describe well what are the requirements for calibration accuracy in practice. The authors only briefly mention possible sources of error in Doppler wind lidars, but do not give an idea about the quantity of such errors. How accurate do lidar measurements have to be for the industry or for research purposes? Maybe the calibration accuracy here is good enough after all? In fact, it cannot be known from this experiment if differences between wheel speed estimation through the calibration and reference wheel speed are due to inaccuracy in the calibration setup or the lidar itself.

We have added to the introduction information about current practice for lidar calibrations, uncertainty of these calibrations and the targeted accuracy of the calibration procedure presented in this manuscript to better motivate the work:

'Current practice for calibrating wind lidars is to use cup or sonic anemometers (Courtney, 2013) as reference instrument and the calibration is often limited by the uncertainty of the reference instrument. Even when using sonic anemometers as reference the overall lidar calibration uncertainty is typically of the order of \$1-2\%\$ (Wagenaar, 2016). This does not do justice to the lidars. We believe that the potential accuracy of wind lidars is much higher than that, lasers are after all very stable instruments and frequency analysis not necessarily faulty, and therefore we propose a new calibration method with a targeted standard uncertainty of \$0.1\%\$. The result of such an order of magnitude increase in accuracy can hopefully propagate through the wind energy industry as higher accuracy will have significant economic benefits.'

In Sect. 4.1 the uncertainty on the reference speed is estimated to 0.02% which is about a fifth of the total estimated uncertainty, but it is correct that for the overall calibration the lidar uncertainty cannot be distinguished from the rig uncertainty. A calibration will always have an uncertainty and this will be the lowest uncertainty we can ever ascribe to the instrument under test. Therefore we believe that lowering the calibration uncertainty by an order of magnitude can be of importance to the use of wind lidars in the wind industry where people are often looking for gaining a fraction of a percent.

• Until the last sentence of the discussion Section 6 I was wondering if it was not a possibility to measure from zero wind speed perpendicular to the wheel surface to get a more continuous calibration curve. It is a bit unsatisfactory and hard to understand why the tilt angle of the lidar cannot be mechanically adjusted for that.

Yes, it is indeed a bit unsatisfactory. Please see our response to Comment "p.10, l.10f" below.

0.2 Specific comments

• Title: The title is a little bit misleading, because I was expecting to see results of the calibration of the lidar, but actually there are only evaluations about the accuracy of the calibration itself, not the lidar.

Yes, the title could perhaps be more specific although we do not find it directly misleading because what we present is actually a calibration of a wind lidar and the associated uncertainty.

**• Introduction:**

- p.1,l.20: I think it would be worthwhile to expand on the kind of laser errors that can cause errors of line-of-sight velocity to better motivate the work.

We do not think it is very relevant to list different sources of errors. The point we are trying to make is that without a calibration (comparing against a known reference) we cannot know if the lidar measures correctly. The reason for any inaccuracy is of course very relevant for the lidar manufacturer but for the operator the main concern is if the lidar measures as it should.

- p.2, l.1: Shinder et al. 2013 should be inside the parantheses. The citation style should be checked throughout the manuscript.

**All citations have been updated**

• Section 2:

 – p.2,I.22ff: Since this is a commercial product, I think it would be sufficient to name the manufacturer and give the accuracy of rotation speed measurement. Measurement principle is not so important for this study.

**We have shortened the paragraph leaving out technical details about the measuring ring works (P3L5-9)**

- Table 1: I recommend to divide the table in two, separating calibration rig properties and lidar properties.

**Table has been divided into two as suggested**

• Section 3:

– All equations: I suggest to use  $\varphi$  (varphi) to be consistent with the plots.

We have updated the figures such that the same symbol,  $\phi$ , is used as in the text.

- p.4, l.2: Just a suggestion for wording: I do not think it is a severe problem, just a technical one.

**Wording changed to:**

'... paradox since for the lidar to measure the true tangential wheel speed, and only that, the overlap between laser beam and wheel surface will need to be infinitely small. In this case there will be no backscatter signal to detect! In order to circumvent this paradox it is therefore necessary to measure a different component of the reference speed than the tangential together with the corresponding tilt angle and from that calculate the measured tangential speed.'

- p.4, l.7: Please expand why the resolution of the Doppler spectrum is a problem. It does not define the resolution of the velocity measurement, because the typical MLEs interpolate the spectrum to

estimate the Doppler shift. I think the main benefit of several tilt angles / wheel speeds is that the tangential speed can then be extrapolated!?

In the case of a very narrow Doppler spectrum e.g. because the probed speed is extremely uniform the spectral resolution becomes an issue. This case actually occurs in our measurements, when the beam is very narrow and when measuring very close to the wheel top and we therefore think it is worth mentioning. However, we agree that the main benefit is that the speed can be extrapolated.

**We have expanded the text:**

'Another source of uncertainty, present at any tilt angle, is the speed estimation uncertainty due to the finite resolution of the measured Doppler spectrum which is especially pronounced when using a very narrow laser beam. The narrow beam leads to only a very limited range of projected speeds being sensed confining the Doppler signal to a single spectral bin. However, this uncertainty can be eliminated by scanning over a range of tilt angles or alternatively, a range of wheel speeds.'

- p.4, l.24: I would not call it a trick, it is geometry after all.

**'trick' changed to 'approach'**

- p.6, l.8f: I think the connection between L'Hospital and the following Taylor's expansion is not made very clear. Maybe Eqs. 9-10 could even be put in the appendix and just be referenced at this point to make it easier to follow the main storyline of the paper.

**Agree. Equations 9-10 and accompanying text have been put into Appendix A**

- p.7, l.11: Why does part of the beam even have to go above the wheel? Wouldn't it make it easier to just choose tilt angles where the whole beam is on the wheel?

Yes it would, but then we wouldn't know the angle between beam and wheel. The main reason for letting the beam skim above the wheel is to establish as accurately as possible the tilt angle corresponding to a true tangential measurement. We have tried to explain this in Sect. 3.6 and Sect. 3.2.1 where we have reformulated to beginning:

'As mentioned above, Eq. (6) only applies as long as all of the beam is on the wheel but if parts of the beam go above the wheel the relationship between  $\frac{V_{text}UOS}{V_{text}UOS}^{J}$  and  $\frac{1}{V_{text}}$  and  $\frac{1}{V_{text}}$  changes and this is unavoidably what will happen when we try to measure as close as possible to the true tangential speed. It is therefore interesting to take a closer look at the special case characterised by  $\frac{1}{V_{text}}$  that is for so small tilt angles that only parts of the beam hit the wheel. Furthermore, measuring in this angular range can be used to estimate the beam width which will be explained in Sect. 3.6:'

- p.9, Eq.17: I think the whole derivation does not have to appear here, especially since the 3D model does not have a significant benefit over the 2D model. It could still be put in the appendix.

**Agree. Derivation of Eq. 15 has been put into Appendix B and only the result is left in the main manuscript**

- p.10, l.10f: Could the beam width maybe be estimated from the tilt angle sweep if it is done so far until none of the beam hits the wheel any more? Or if instead of trying to hit the tangential, the beam was set to hit the wheel frontal, which would lead to zero velocity measurement?

Yes indeed, and that is what we do and have tried to explain in Sect. 3.6.

Regarding measuring close to perpendicular to the wheel surface, then it is definitely an idea that is worth considering for a future study, but it will require a redesign of the calibration rig (the beam can simply not be tilted enough with the present design) and it has not been possible to perform such a study yet.

- p.12, l.11f: It is not clear to me why the extrapolation from angles larger  $\theta$ 1 is not the real tangential speed ratio (bi = 1!?).

The finite beam width introduces an offset such that the regime where the speed ratio falls off as - L/R is shifted towards higher tilt angles and an extrapolation from this regime will overestimate the tangential speed. We have tried to illustrate this in Fig. 7 (b) where the dashed line represents the extrapolation.

• Section 4:

- p.14, Eq.27: It would be good to introduce a variable for VLOS Vwheel and use this as subtext in this equation.

Symbol  $\Lambda$  introduced to denote the ratio Vlos/Vwheel.

• Section 5:

– p.16, l.26: "quite well", "some of the truth" and "at least relatively" are such weak statements that
 I strongly recommend to reformulate.

**Wording changed to:**

'The first thing to notice is the clear resemblance in the shape of the two curves indicating that this is a valid method for estimating the beam width although the absolute values do not agree. Actually, the values of the calculated widths are about three times higher than the measured, but this is not too disturbing since we are not expecting the measured width to represent the \$1/e^2\$-width but rather the width from where we can detect a signal.'

- p.20, l.18: %/°is not a good notation and it is also mixed with %/°.

**Wording changed to: "percent per degree"**

- p.23, l.2ff: I was wondering about this possibility throughout the whole derivation of the calibration models for  $\theta < \theta 1$ . Maybe it should be explained earlier, why it is not possible.

As mentioned above this is more a practical issue that our setup does not allow for such a study and we prefer to keep the short explanation in the discussion. However, we have added a small comment about an implication we see regarding the uncertainty:

'In this case the calibration uncertainty would depend critically on the angle measurement uncertainty'

- p.23, l.25: "its" instead of "it's".

**Corected**

- p.23, l.19f: Is it enough to calibrate continuous wave lidars at a minimised beam width? What would an industrial calibration have to cover?

We believe that it is sufficient to calibrate at a minimised beam width. Best practice is to use the lowest possible calibration uncertainty which achieved with a minimised beam width and

furthermore we see no reason why the actual lidar uncertainty should change with beam width. We have changed 'uncertainty' to 'calibration uncertainty'.

**Reviewer #2**

**General comments**

The paper of Pedersen and Courtney is well structured and written. They are thorough in their way of deriving models, uncertainties and finally comparing it to measurements. Their approach to use a flywheel in combination with a Doppler wind lidar seems new and worth publishing (after minor revisions).

**Specific comments**

Concerning the estimate of  $\theta$ 1 (P12L5-6 and P16L18-21). Did you also measure backscatter? Why not look at the plot of tilt vs backscatter? I suppose it should be increasing from  $\theta$ 0 to  $\theta$ 1 strongly and afterwards only slightly (if at all)? If it turns out that the increase in backscatter shows no point of change around  $\theta$ 1, just make a different set up where you place something (with a sharp edge) on top of the (not turning) wheel and tilt and measure the backscatter as the beam hits the object first partially and then completely. Now the plot should just show an increase in backscatter from the angle where the beam touches the object partially to the angle where it hits it completely and after that stay constant? The difference could be an estimate for  $\Delta\theta$ ? (To avoid backscatter from walls, use window or something that reflects at an angle as background?). If my thoughts on this are correct but a new setup/measurement is too time consuming please address/discuss this appropriately in the document.

The thoughts are correct and definitely a good idea! Unfortunately, our lidar does not measure the backscatter and we cannot perform the suggested analysis on the available data and reprogramming the lidar and repeating the measurements would be too time consuming. Instead we have added a paragraph to the discussion addressing this issue:

'The uncertainty analysis shows that the main uncertainty contributor is  $U_{\$  Delta\theta}\$ which essentially depends on the beam width estimation. An estimate for  $\Delta = 0$  and theta angles for the first sporadic and the first stable measurements, respectively, was proposed. Another approach that could potentially reduce this uncertainty is to measure the backscatter level as function of tilt angle. In this way the backscatter signal would increase strongly from  $\theta = 0$ theta\_0\$ to  $\theta = 0$  and wheel increases and then remain more or less constant when the entire beam is on the beam. Unfortunately, our lidar in its present state does not measure or store the backscatter level and therefore this approach has not been tested.'

In section 3.1 you describe your approach to rotate around the transceiver lens for the model. But I did not see you describing where the tilt-axis lies relative to the telescope for your measurements. If you raise/lower just the end so it rotates around its lens (As the caption of Fig.2 suggests), please include this information somewhere. If the tilt axis is not going through the lens please also address this (maybe add some text that makes clear that the changes/differences to L and yr are negligible).

The telescope tilts around a point located about 10 cm directly below the lens. This information is added to Sect. 2.

The influence of this is negligible and a comment about this added to Sect. 3.1:

'As mentioned in Sect. 2 the physical beam actually rotates around a point situated beneath the lens but for a tilt angle of 2.5°, 5 which is the maximum attainable in our setup, this leads to a negligible difference for xr and yr of about 4 mmand 0.1 mm, respectively'

Although it is obvious that no wind speeds are measured because the title says "flywheel calibration"- It is after all a "Doppler wind lidar". As someone who uses a Doppler wind lidar to measure wind speeds it feels a bit weird to read the whole paper and end up just with a calibration "for rotating steel". I of course prefer a lidar that goes through such a quality check, but still... the journal is ATMOSPHERIC Measurement Techniques... Maybe you could add a paragraph about how and why this translates to wind measurements or state that this calibration is meant more as a necessity/possibility than as a sufficiency for Doppler wind lidars quality? (This may be a matter of taste... if you feel all is clear by using the word "calibration" that is also fine)

We have added a paragraph to the introduction addressing this subject:

'It might seem strange to use a rotating steel wheel as measurement target, after-all the lidar is intended for measuring on small aerosols carried by the wind and not a solid metal target. On the other side, the lidar fundamentally measures a frequency shift in the backscattered light due to a relative motion and it is this frequency shift measurement and the subsequent conversion to a speed we wish to calibrate. The origin of the backscatter is in this connection of less importance. One could, however, envision a scenario where a lidar calibrated in this fashion is used to calibrate another lidar in e.g. a wind tunnel or the free atmosphere. This would lead to a calibration procedure resembling that of the current practice but where the limiting accuracy of a cup anemometer is alleviated.'

**Technical corrections**

Code and data availability: Pedersen, A. T.: Flywheel calibration of coherent Doppler wind lidar - data, https://doi.org/10.11583/DTU.11991189, 2020.

gives me -> Page not found

**Our university library service has been advised about this and data will be available shortly.**

https://amt.copernicus.org/preprints/amt-2020-88/ says "Anders Tegtmeier Pedersen and Pedersen Courtney" and the paper itself says "Anders Tegtmeier Pedersen and Michael Courtney"

**We will contact AMT about this.**

There are a bunch of "r"-index is missing: EQ9, P6L15-16, EQ10, P7L5/6/9/11 In the text and figures, you use different styles of the Greek phi ( $\phi/\phi$ ). Please make this consistent.

**Equations have been corrected and figures updated such that $\phi$ is used throughout.**

You use "best" 5 times in the document. When you say "our best" I get it, but just "best" is a bit bold. Maybe rephrase some occurrences.

**Wording changed to 'our best'**

P2L19-20 The reference to Fig.1 makes it seem like we should be able to see the inclinometer on top of the telescope. I don't see it... maybe use labels / zoom in the Fig.1 or if it is an old picture without inclinometer move the Fig.1 reference to one sentence earlier?

Yes, it is an old photo. Figure reference moved one sentence back.

**Added to Fig. 1 caption: 'Inclinometer not visible in the photo'**

P6L8 "this" is ambiguous. Maybe use "the right hand side divided by Vwheel"

Wording changed: '...the right hand side divided by ...'

EQ8  $\theta$  should not be there, right?

No. Removed.

P6L15-16 not listed as equations

**Corrected**

P8L15 should be  $yr - R \le y \le yr + R$  or  $-R \le y - yr \le R$ , right?

Yes. Manuscript updated.

EQ17  $\varphi$  is missing "r"-index EQ17 I don0t follow the last equal sign. Please explain, expand or correct.

**Derivation expanded**

P10L4 "as long the" -> "as long as the"

Corrected

P11L14 "cause" -> "causes"

Corrected

P12L10 "arise" -> "arises"

Corrected

P13L8 "wee" -> "we"

Corrected

P14L5 "the are" -> "there are"

Corrected

P14L19 "shown i the" -> "shown in the"

**Corrected**

P11L18-20 Please check the sentence structure again. The last bit doesn't seem right. May be "way than"->"way other than". Maybe even rephrase as "direct angle measurement" is ambiguous. Did you mean "direct angle measurement from inclinometer"?

Wording changed to: '... other than a direct angle measurement with the inclinometer'

P15L8 "assume" -> "assumed"

Corrected

EQ38 index cc should be bc

Corrected

EQ39 I don0t follow this transformation. Did you swap "wheel" and "LOS"?

Yes. Corrected

P17L3 "0.14m" -> "0.14 mm"

Corrected

P17L7 "Table2" -> "Table 2"

Corrected

P17L15-17 Sentence looks wrong. Maybe "measurement widening" -> "measurement is widening"?

Wording changed to: '...is covered in each measurement which spreads the Doppler signal over several bins'

P19L4 "shape curve" -> "shape the curve"

**Corrected**

**Cosmetic suggestions (These need not be address)**

P5L4 "rearranging and inserting into Eq. (1)" with the page break - I found it hard to follow (first time reading), that you also use eq.3 in this step.

Fig.6 An arrow for wRe cos(phir) could be nice, but I guess it overlaps with the redline? (Maybe dashed or dotted arrow?)

**Figure updated**

P13L1 double usage of "end" is hard to read

Sentence rephraised:

'In the end we therefore arrive at an estimate of the ratio...'

P15L22-26 I would rephrase it... explain/motivate uθw differently.

P17L6 "relative uncertainty of" -> "relative uncertainty uθw of"

P17L7 "absolute uncertainties can" -> "absolute uncertainties  $u\Delta\theta$  can"

Fig.9/10 different colour for residuals lines?

[revised manuscript text omitted]
_{\text{LOS}} = V_{\text{wheel}} \cos(\phi_{\text{s}} + \theta) = \omega R \cos(\phi_{\text{s}} + \theta), \tag{1}$$

where  $V_{\text{wheel}}$  is the peripheral speed,  $\phi_s$  is the skimming angle,  $\theta$  is the beam tilt angle,  $\omega$  the angular frequency, and R the radius of the wheel.

15

Now, to find the relation between  $\phi_s$  and  $\theta$  we can make use of a little trick an approach which will also prove valuable later on; instead of tilting the beam we rotate the centre of the wheel,  $(x_0, y_0)$ , an angle  $\theta$  around the centre of the transceiver lens which defines the origo of our coordinate system, see Fig. 3. The new centre of the wheel is denoted  $(x_r, y_r)$ . From the figure it is clear that the angle,  $\phi_r$ , between the vertical and the intersection point between beam and wheel is

$$\phi_r = \phi_s + \theta,\tag{2}$$

and that

10

$$\cos(\phi_r) = \frac{-y_r}{R}.$$
(3)

From Fig. 2 we see that  $(x_0, y_0) = (L, -R)$  and from this we can calculate  $(x_r, y_r)$  via the rotation matrix  $\underline{\underline{R}}_z(\theta)$

$$\begin{pmatrix} x_r \\ y_r \end{pmatrix} = \underline{\underline{R}}_z(\theta) \begin{pmatrix} x_0 \\ y_0 \end{pmatrix} = \begin{pmatrix} \cos\theta & -\sin\theta \\ \sin\theta & \cos\theta \end{pmatrix} \begin{pmatrix} L \\ -R \end{pmatrix} = \begin{pmatrix} L\cos\theta + R\sin\theta \\ L\sin\theta - R\cos\theta \end{pmatrix}.$$
(4)

5 As mentioned in Sect. 2 the physical beam actually rotates around a point situated beneath the lens but for a tilt angle of  $2.5^{\circ}$ , which is the maximum attainable in our setup, this leads to a negligible difference for  $x_r$  and  $y_r$  of about 4 mm and 0.1 mm, respectively. Now, rearranging and inserting into Eq. (1) we finally arrive at

$$\frac{V_{\rm LOS}}{V_{\rm wheel}} = \frac{R\cos\theta - L\sin\theta}{R}.$$
(5)

In our setup  $\theta$  is small reaching a maximum value of Since the maximum tilt angle is only about 2.5° so we can make the approximations that  $\cos \theta = 1$  and  $\sin \theta = \theta$  such that

$$\frac{V_{\rm LOS}}{V_{\rm wheel}} \approx 1 - \frac{L\theta}{R}.$$
(6)

It is thus seen that for small tilt angles there is a linear relationship between  $\frac{V_{LOS}/V_{wheel}}{V_{wheel}}$  the speed ratio  $\frac{V_{LOS}}{V_{wheel}}$  and  $\theta$  and this can be utilised in the calibration procedure.

**15 3.2 Finite width, collimated beam, 2D**

Now, a real laser beam is of course not infinitely narrow but has a transverse profile of finite width, e.g. the laser used in this study has a Gaussian profile. We therefore expand the model to include the beam width radius, w, but to begin with limit our

selves to the two dimensional case and the assumption that the beam intensity has a constant transverse cross-section, i.e. the intensity profile across the beam has a "top hat shape".

For a beam of finite width a finite part of the wheel perimeter will be illuminated by the laser and thus a range of line-of-sight speeds be measured, see Fig. 4. Each incremental line-of-sight speed will be

5
$$dV_{\text{LOS}} = V_{\text{wheel}} \cos(\phi_r),$$
 (7)

and these will each contribute a proportion  $d\phi_r/\Delta\phi_r$  of the total speed sensed by the lidar, where  $\Delta\phi_r (= \phi_{r_1} - \phi_{r_0}) \Delta\phi_r = \phi_{r_1} - \phi_{r_0}$ is the total angle subtended by lidar illumination. The total speed contribution  $V_{\text{LOS}}$  is thus obtained by integrating Eq. (7) whilst normalising by  $\Delta\phi_r$

$$V_{\text{LOS}} = \frac{1}{\Delta\phi_r} \int_{\phi_{r_0}}^{\phi_{r_1}} V_{\text{wheel}} \cos\phi \underline{+\theta} d\phi = \frac{1}{\Delta\phi_r} V_{\text{wheel}} \left(\sin\phi_{r_1} - \sin\phi_{r_0}\right).$$
(8)

10 By applying L'Hospital's rulethis, the right hand side divided by  $V_{wheel}$  is easily seen to reduce to  $\cos \phi_r$  as  $\phi_{r_1}$  approaches  $\phi_{r_0}$ , i.e. as the beam becomes narrower, and therefore give the same result as in Sect.3.1.

If we apply Taylor's expansion to the third order to Eq. we get

$$\begin{array}{lcl} \displaystyle \frac{V_{\rm LOS}}{V_{\rm wheel}} & \equiv & \displaystyle \frac{1}{\Delta\phi} \left(\phi_1 - \frac{1}{6}\phi_1^3 - \phi_0 + \frac{1}{6}\phi_0^3\right) \\ \\ & \equiv & \displaystyle \frac{1}{\Delta\phi} \left(\Delta\phi - \frac{1}{6}\left(\phi_1^3 - \phi_0^3\right)\right) \\ \\ & \equiv & \displaystyle 1 - \frac{1}{6}(\phi_1^2 + \phi_0^2 + \phi_1\phi_0), \end{array}$$

15

and if we further make the approximations-

$$\begin{array}{rcl} \underline{\phi_0} & \equiv & \underline{\phi_m - \delta}, \\ \\ \phi_1 & \equiv & \phi_m + \delta, \end{array}$$

where  $\phi_{r_m}$  is the mean of  $\phi_{r_0}$  and  $\phi_{r_1}$  and  $\delta$  is a small perturbation we get

20
$$\frac{V_{\text{LOS}}}{V_{\text{wheel}}} = 1 - \frac{1}{6}(\phi_1^2 + \phi_0^2 + \phi_1\phi_0) \equiv \frac{1 - \frac{1}{6}(3\phi_m^2 + \delta^2) \approx 1 - \frac{1}{2}\phi_m^2 \approx \cos\phi_{r_m}}{1 - \frac{1}{6}(3\phi_m^2 + \delta^2) \approx 1 - \frac{1}{2}\phi_m^2 \approx \cos\phi_{r_m}}$$

which is seen to be equal to Eq. -3.1 and the model is thus mathematically consistent with the 1D model.

This means that even Even for a beam of finite width Eq. (6) is a reasonable good approximation to how the ratio  $\frac{V_{LOS}}{V_{Wheel}}$  changes as the beam is tilted. This can For completeness, a mathematical derivation of this is presented in Appendix. A but from a physical point of view it can intuitively be understood as that the high speed measured at  $\phi_1$  is more or less balanced

by the low speed measured at  $\phi_0$ . However, the approximation only applies as long as the entire beam cross-section is on the wheel; if part of the beam goes above the wheel, as it will for very small tilt angles, the relationship changes as we shall see in Sect. 3.2.1.